# Conformational changes in Lassa virus L protein associated with promoter binding and RNA synthesis activity

Tomas Kouba [1,4], Dominik Vogel [2,4], Sigurdur R. Thorkelsson [3,4], Emmanuelle R. J. Quemin [3], Harry M. Williams [2], Morlin Milewski [2], Carola Busch[2], Stephan Günther [2], Kay Grünewald [3], Maria Rosenthal [2,5✉] & Stephen Cusack [1,5✉]

Lassa virus is endemic in West Africa and can cause severe hemorrhagic fever. The viral L protein transcribes and replicates the RNA genome via its RNA-dependent RNA polymerase activity. Here, we present nine cryo-EM structures of the L protein in the apo-, promoter-bound pre-initiation and active RNA synthesis states. We characterize distinct binding pockets for the conserved 3' and 5' promoter RNAs and show how full-promoter binding induces a distinct pre-initiation conformation. In the apo- and early elongation states, the endonuclease is inhibited by two distinct L protein peptides, whereas in the pre-initiation state it is uninhibited. In the early elongation state, a template-product duplex is bound in the active site cavity together with an incoming non-hydrolysable nucleotide and the full C-terminal region of the L protein, including the putative cap-binding domain, is well-ordered. These data advance our mechanistic understanding of how this flexible and multifunctional molecular machine is activated.

[1] European Molecular Biology Laboratory, Grenoble, France. [2] Bernhard Nocht Institute for Tropical Medicine, Hamburg, Germany. [3] Centre for Structural Systems Biology, Leibniz Institute for Experimental Virology, University of Hamburg, Hamburg, Germany. [4] These authors contributed equally: Tomas Kouba, Dominik Vogel, Sigurdur R. Thorkelsson. [5] These authors jointly supervised this work: Maria Rosenthal, Stephen Cusack. ✉email: rosenthal@bnitm.de; cusack@embl.fr

assa virus (LASV) is a segmented, negative-strand RNA virus belonging to the family of *Arenaviridae* within the *Bunyavirales* order. It is a rodent-borne virus, endemic to West Africa and the causative agent of Lassa haemorrhagic fever, a febrile illness with increasing case numbers and a case fatality rate among hospitalised patients of ~15% in Nigeria in 2018[1]. Recent studies applying computational modelling predict a total number of ~900,000 human infections per year across West Africa[2]. The large (L) protein of LASV is a multi-domain molecular machine that binds the conserved 3′ and 5′ ends (the 'promoter') of each of the two viral RNA (vRNA) genome segments (denoted L and S) and plays a central role in the viral life cycle, which is entirely cytoplasmic. The L protein contains an RNA-dependent RNA polymerase (RdRp) activity and catalyses both viral transcription and genome replication. Each vRNA segment is a template for the synthesis of two different types of RNA products: capped viral transcripts (of L and NP genes) as well as an unmodified full-length complementary RNA (cRNA) copy, which is an intermediate of viral genome replication. The cRNA is a template for the second stage of replication, the synthesis of vRNA genome copies, as well as the production of further mRNA (of GPC and Z genes) by transcription. Transcription is initiated using a capped primer derived from host mRNA by a yet to be elucidated 'cap-snatching' mechanism involving the intrinsic endonuclease (EN) of the L protein and possibly its cap-binding domain (CBD)[3]. This results in viral mRNAs that have 1–7 host-derived nucleotides at the 5′ end[4–6]. Viral genome replication is initiated by a prime-and-realign mechanism resulting in an extra G nucleotide at the 5′ end of the vRNA and cRNA[7]. The first structural studies of the complete arenavirus L protein were conducted on Machupo virus (MACV), which is related to LASV but belongs to the group of New World arenaviruses found in South America. Negative stain electron microscopy studies at low resolution revealed a donut-like molecule with accessory appendages[8]. In 2020, the first models of MACV and LASV L proteins were proposed based on cryo-electron microscopy (cryo-EM) with overall resolutions of ~3.6 and 3.9 Å, respectively[9]. These structures revealed that arenavirus L proteins are structurally similar to the polymerases of the related La Crosse and influenza viruses[10–14]. However, the reported arenavirus structures are incomplete and do not show the L protein in an active conformation[9].

Here, we present nine cryo-EM structures that provide insights into the conformational rearrangements of LASV L protein that occur upon its activation into a functional RNA synthesis elongation state. This comprehensive structural study is complemented by biochemical data from in vitro assays with purified L protein and selected mutants as well as functional data in cells using the LASV mini-replicon system[15]. The results presented enhance our mechanistic understanding of the multifunctional LASV L protein and will guide targeted drug development approaches in the future.

## Results

**Overview of structures obtained.** Nine cryo-EM structures of LASV L protein have been determined in the apo-state, bound to the 3′ end of the genomic vRNA alone, bound to the full promoter, comprising the highly complementary 3′ and 5′ ends of the vRNA, or in a stalled, early elongation state (Table 1).

From grids of the apo-state, two different 3D classes were separated. In the APO-ENDO structure, at 3.35 Å resolution (Fig. 1a), the N-terminal EN is clearly resolved, packing against the polymerase core and with a peptide (residues 1092–1104) from the central region of the L protein bound in its active site cavity, presumably inhibiting its activity. In the second class, denoted APO-RIBBON (Fig. 1a), at 3.73 Å resolution, the EN is not resolved, but residues 822–1110 of the L protein form an extended structure including an α-bundle (α-ribbon, 843–884, with a third helix 907–925 packing against it) that is not visible in the APO-ENDO structure. Masked refinement of the common regions of the two apo-structures yielded a map of the APO-CORE (Fig. 1a) with an improved resolution of 3.14 Å that allowed a more accurate model to be built.

Upon incubation of L protein with nucleotides (nts) 1–16 of the vRNA 3′ end alone (structures denoted 3END-CORE, 2.70 Å, 3END-ENDO, 3.04 Å) (Fig. 1b), nucleotides 1–6 from the 3′ end bind specifically in a buried groove under the pyramid, a prominent feature in the N-terminal region of the L protein

---

**Table 1 Overview of the different LASV L protein structures.**

| Identifier | Resolution [Å] | RNA ligands | PDB accession | EMDB accession | Comment |
|---|---|---|---|---|---|
| APO-CORE | 3.14 | - | 7OCH | 12807 | High-resolution apo-polymerase core, best defined Zn$^{2+}$ coordination site |
| APO-ENDO | 3.35 | - | 7OE3 | 12860 | EN domain complete and bound to inhibitory peptide 1092–1104, well-defined interaction with polymerase core |
| APO-RIBBON | 3.73 | - | 7OE7 | 12953 | α-ribbon visible |
| 3END-CORE | 2.70 | 3′ 1–16 | 7OEA | 12862 | Highest resolution core, 3′ RNA bound to the secondary binding site, EN bound to inhibitory peptide 1092–1104 |
| 3END-ENDO | 3.04 | 3′ 1–16 | 7OEB | 12863 | EN domain complete and bound to inhibitory peptide 1092–1104, 3′ RNA bound to the secondary binding site |
| PRE-INITIATION | 3.34 | 5′ 0–19 3′ 1–19 | 7OJL | 12955 | dsRNA promoter bound, α-bundle and pendant visible |
| MID-LINK | 3.50 | 5′ 10–19 3′ 1–19 | 7OJJ | 12861 | C terminus visible (low resolution), EN not potentially autoinhibited |
| DISTAL-PROMOTER | 3.89 | 5′ 10–19 3′ 1–19 | 7OJK | 12954 | Distal dsRNA promoter bound, α-bundle and pendant visible |
| ELONGATION | 2.92 | 5′ 0–19 3′ 1–19 C8 primer, UMPNPP | 7OJN | 12956 | C terminus buildable with high confidence, RNA duplex in active site, UMPNPP, EN autoinhibited by peptide 173–190 |

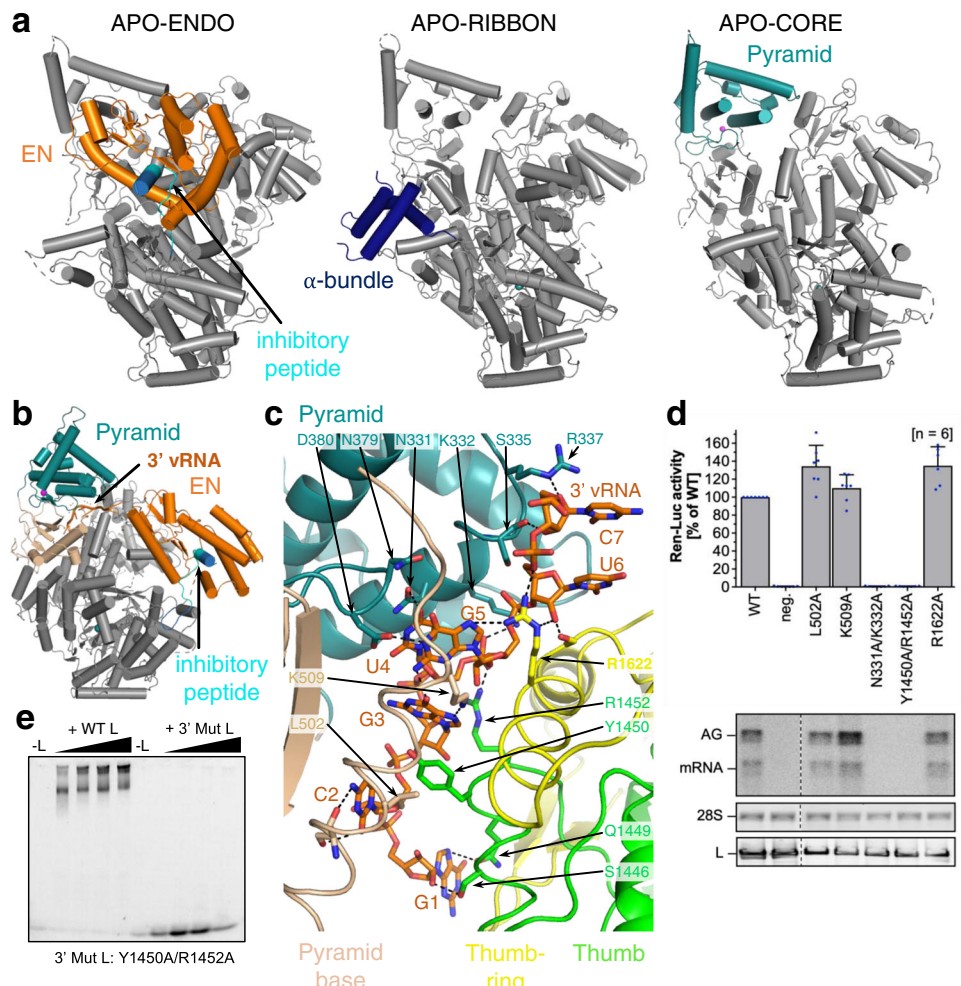

**Fig. 1 L protein in the apo-state and with 3′ viral RNA bound in the secondary binding site. a** Ribbon diagram presentations of the structures APO-ENDO, APO-RIBBON and APO-CORE. Each of the respective experimental maps resolves distinct regions of the L protein better than the others and those regions are shown in colour and indicated by name in the respective structures. **b** Overall structure of L protein 3END-CORE as a ribbon diagram with the 3′ vRNA bound below the pyramid domain. Pyramid (teal), pyramid base (wheat), EN domain (orange) and the inhibitory peptide (cyan) are highlighted. **c** Close-up of the secondary 3′ vRNA binding site with the 3′ vRNA nucleotides 1–7 (orange), pyramid domain (teal), pyramid base (wheat) as well as thumb (green) and thumb-ring (yellow) domains. Important amino acids in the RNA-protein interface are shown as sticks with respective labels. Hydrogen bonds are indicated by dotted lines. For selected regions, secondary structure depiction was disabled to enhance visibility. **d** LASV mini-replicon data for L proteins with mutations in the secondary 3′ RNA binding site presenting luciferase reporter activity (in standardised relative light units relative to the wild-type L protein (WT)). Data were presented as mean values ± SD of at least six biological replicates (n = 6), although for most mutants seven biological replicates were included. All biological replicates are shown as blue dots (top panel). Middle panels present Northern blotting results with signals for antigenomic viral RNA (AG), viral mRNA (mRNA) and 28 S ribosomal RNA (28 S) as a loading control, and the bottom panel shows Western blot detection of FLAG-tagged L proteins (L) to demonstrate general expressibility of the mutants. Source data are provided in a Source Data file. **e** Electrophoretic mobility shift assay of wild-type L protein (WT L) and mutant Y1450A/R1452A (Mut L) with 10 nt 3′ viral RNA. L protein concentrations ranging from 0–1 μM and 0.2 μM of fluorescently labelled 3′ vRNA (Supplementary Table 1) were used (see methods).

(Fig. 1c). This site corresponds to the secondary 3′ end-binding site previously described for influenza virus, La Crosse virus (LACV) and MACV polymerase proteins[9,13,16–18]. In these structures, the EN remains in the inhibited conformation as observed in the APO-ENDO structure (compare Fig. 1a and b). When the full vRNA promoter is bound (5′ end nts 0–19, including an additional G0 according to the product expected from the prime-and-realign initiation mechanism, 3′ end nts 1–19), a pre-initiation complex (PRE-INITIATION) is observed at 3.34 Å resolution (Fig. 2a, Supplementary Fig. 1). This structure reveals that the LASV vRNA promoter is organised similarly to those of influenza virus[10] and LACV[13,14] in comprising a single-stranded 5′ end folded as a hook, a distal duplex region and a single-stranded 3′ end, only partially visible, directed towards the

RNA synthesis active site (Fig. 2c). Overall, the protein conformation of the PRE-INITIATION structure resembles that of the APO-RIBBON, with an additional partially ordered insertion domain, previously denoted the pendant[9], packing against the 3′ strand of the promoter (see below) (Fig. 2a). Two further structures were obtained from a sample in which the L protein was incubated with a truncated promoter (5′ nts 10–19, 3′ nts 1–19), which lacks nucleotides 0–9 of the 5′ end. One 3D class from this sample, obtained by focussed refinement on the promoter-bound region, (DISTAL-PROMOTER, 3.89 Å resolution), closely resembles the full promoter (PRE-INITIATION) structure, but additionally reveals a new position of the EN, without inhibitory peptide bound (Supplementary Fig. 1). The second 3D class from the same grid (MID-LINK, 3.50 Å

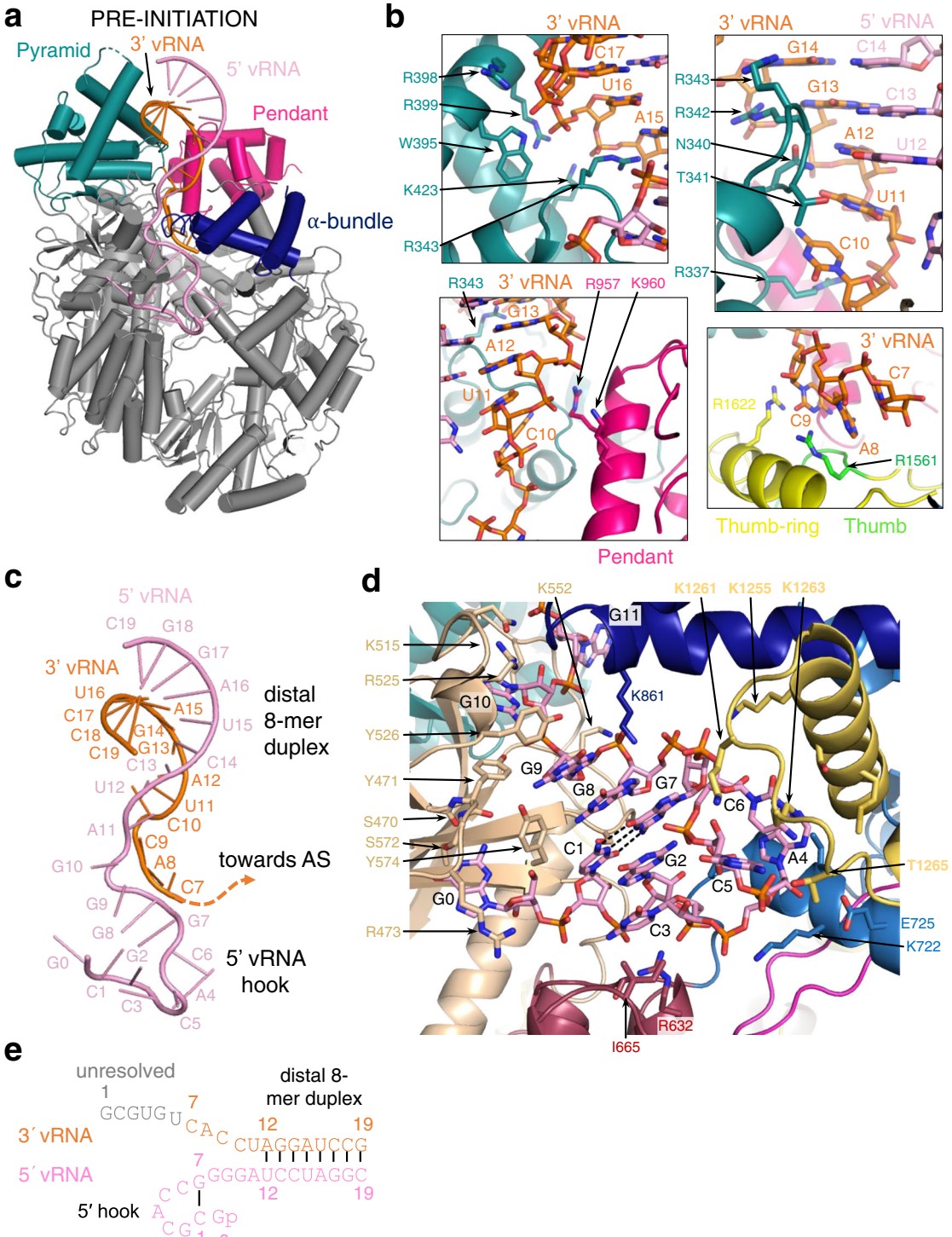

**Fig. 2 L protein in the pre-initiation state. a** Ribbon diagram of the PRE-INITIATION structure with pendant domain (pink), α-bundle (dark blue), pyramid (teal), 3′ vRNA nts 7–19 (orange) and 5′ vRNA nts 0–19 (pink) highlighted in colour and indicated by name. **b** Interactions of the L protein pyramid (teal), pendant (pink) thumb-ring (yellow) and thumb (green) domains with the 3′ vRNA are shown. Important amino acid side chains and the RNA nucleotides of 3′ and 5′ vRNA are shown as sticks with respective labels. **c** Viral RNA observed in this structure with a 5′ vRNA hook structure composed of 5′ vRNA nts 0–9 and a distal duplex region involving 5′ vRNA nts 12–19 and 3′ vRNA nts 12–19. The 3′ vRNA nts 1–11 are directed towards the RdRp active site (towards AS) but not resolved. **d** Close-up of the 5′ RNA hook binding site involving the fingers domain (blue), finger node (light yellow), pyramid base (wheat) and helical region (raspberry). Residues important for the RNA-protein interface and nucleotides are shown as sticks and are labelled. **e** Schematic presentation of the promoter RNA (3′ vRNA in orange, 5′ vRNA in pink) in the PRE-INITIATION structure. Nucleotides 1–6 of the 3′ vRNA, which are not resolved, are coloured in grey. Distal duplex and 5′ hook regions are labelled.

resolution) (Supplementary Fig. 2), obtained by focussed refinement of the other end of the polymerase, shows density for the C-terminal region of the L protein beyond residue 1834. This allows tentative modelling of domains that resemble the mid-link and 627-domains of the influenza virus polymerase PB2 subunit. At very low resolution, an envelope of the putative cap-binding

(CBD-like) domain is observed. A final structure (ELONGATION, 2.92 Å resolution) captures an early elongation state initiated with an uncapped primer and stalled after incorporation of four nucleotides by an incoming non-hydrolysable UTP analogue, UMPNPP (Figs. 3, 4). In this structure, the promoter duplex is disrupted due to translocation of the template and a

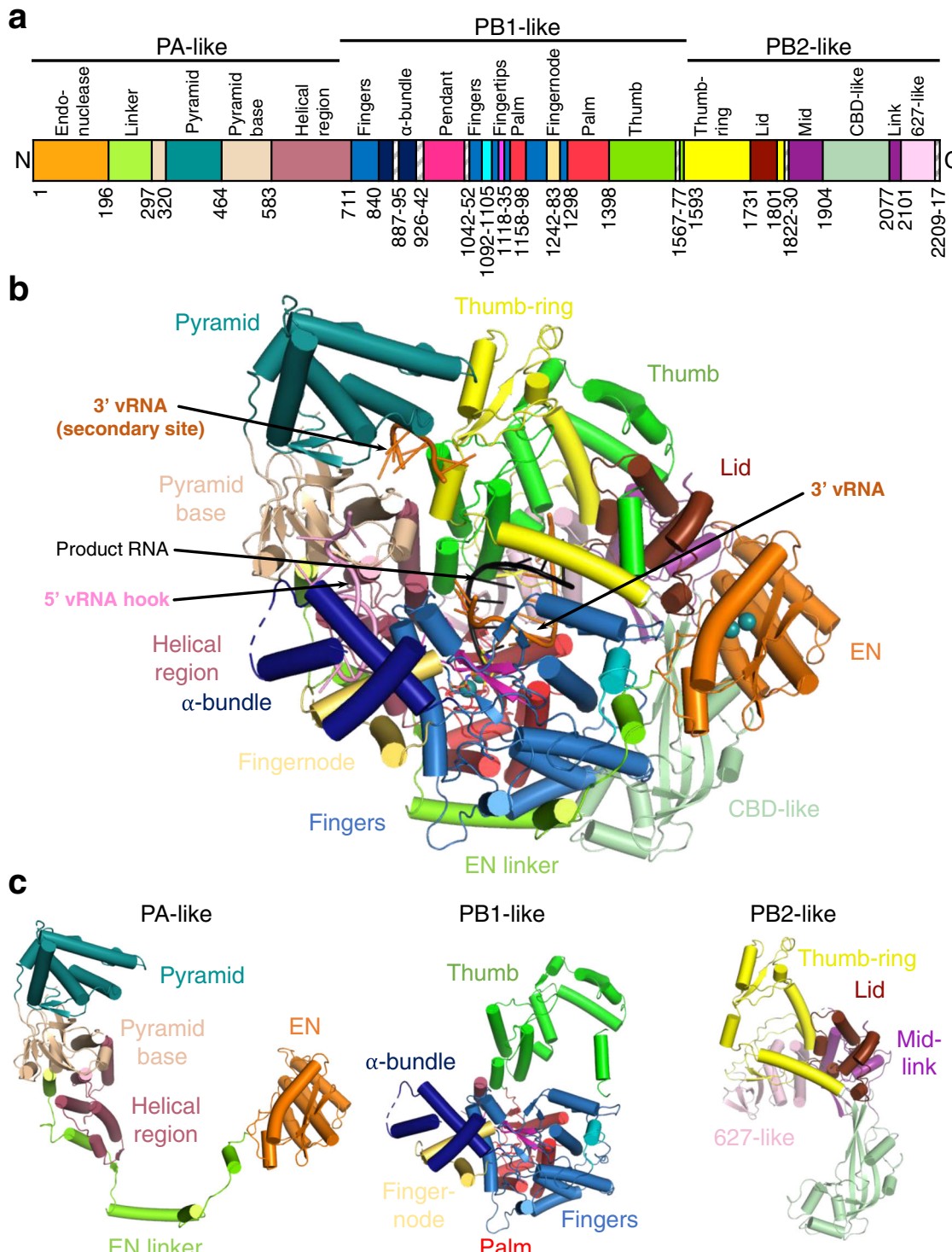

**Fig. 3 Overview of the L protein structure. a** Schematic linear presentation of the L protein domain structure. **b** Complete ELONGATION structure of the L protein presented as a ribbon diagram in front view. Domains are coloured according to (**a**) and labelled. 3' vRNA is coloured in orange, 5' vRNA in pink and product RNA in black. See also Supplementary Movie 3 for a 3D impression of the L protein and its domains. **c** Separate presentation of the PA-like, PB1-like and PB2-like regions of the L protein in the ELONGATION structure.

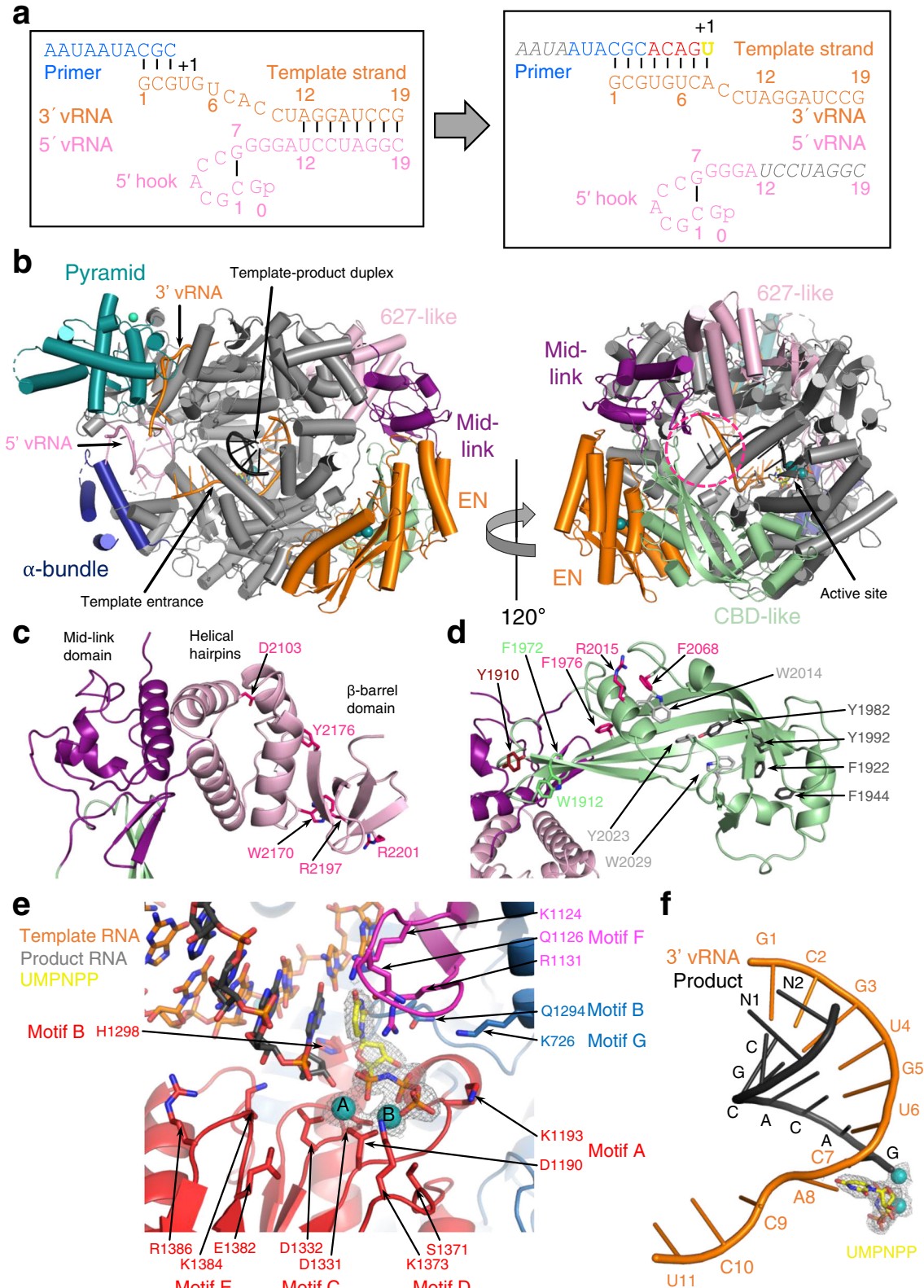

duplex of eight base pairs occupies the active site cavity (Fig. 4a). The complete C-terminal domain is well-resolved, and, due to rotation of the CBD-like domain, is now in a bent configuration, rather than the extended configuration seen in the MID-LINK structure. The C-terminal domain, together with the EN, now in a third distinct position, forms a ring around the putative product exit channel (Fig. 4b).

The different structures obtained (Table 1) reveal that LASV L, similar to influenza virus polymerase[19], has a number of domains flexibly linked to the polymerase core, allowing multiple conformations of the protein. None of the here reported individual structures are complete (although the elongation structure model comprises ~90% of the residues, lacking mainly the pendant domain), but the integration of all the information leads to a coherent picture of the

**Fig. 4 Elongation state of the L protein. a** Schematic presentation of the primed reaction was carried out to obtain the ELONGATION structure with the L protein stalled in an early elongation state. Nucleotides that are not visible or not clearly assignable from the experimental map are shown in grey italics. **b** ELONGATION structure of the L protein presented as a ribbon diagram in two views as indicated. EN, pyramid, α-bundle, mid-link and 627-like domains are coloured. 3′ vRNA is shown in orange, 5′ RNA in pink and product RNA in black. A dashed circle (hot pink) indicates the putative product exit. **c** Close-up on the mid-link and 627-like domains with the respective structural features labelled and the side chains of amino acids shown to be selectively important for viral transcription by Lehmann et al. 2014[24] shown as pink sticks. **d** Close-up on the CBD-like domain with side chains of amino acids that have been tested in the LASV mini-replicon system shown as sticks (pink—selective role in viral transcription; light grey—no significant reduction of L protein activity upon mutation shown by Lehmann et al. 2014[24]; green—no or weak effect on L protein function upon mutation; dark red—potential selective role in viral transcription; dark grey—general defect of L protein activity upon mutation). Corresponding mini-replicon data are presented in Supplementary Fig. 8. **e** Close-up of the polymerase active site with the template RNA (orange), the product RNA (dark grey), the non-hydrolysable UTP (UMPNPP, yellow) and catalytic manganese ions (teal, A and B) involving the palm (red), fingers (blue), fingertips (magenta) and thumb (green) domains of the L protein. Important side chains are shown as sticks and conserved RdRp active site motifs (A–G) are labelled. The map around the UMPNPP and the manganese ions is shown as a grey mesh. **f** Template-product duplex of the polymerase active site is shown as a ribbon diagram with the product in black, the 3′ template in orange, the non-hydrolysable UTP (UMPNPP) in yellow and the catalytic manganese ions as teal spheres. The map around the ions and the UMPNPP is shown as a grey mesh.

overall LASV L protein structure and the significant conformational changes that occur upon promoter binding and the subsequent transition into the active elongation state.

**Overview of LASV L protein structure.** As previously established, MACV and LASV L proteins have an overall architecture similar to previously determined orthomyxovirus and bunyavirus polymerases[9]. With reference to the heterotrimeric influenza virus polymerase, the LASV L protein can be conveniently divided into PA-like (1–687), PB1-like (688–1592) and PB2-like (1593–2217) regions (Fig. 3a, c and Supplementary Fig. 3).

The PA-like region has an N-terminal EN (1–195), whose structure and properties have previously been studied[20,21]. Whereas previous crystal structures of the isolated LASV EN are not resolved beyond residue ~173[20,21], the full L protein structures show that the domain comprises an additional helix ending at 190. The EN is followed by an extended linker (196–296) that wraps around the polymerase core (Fig. 3 and Supplementary Fig. 3). This connects to the pyramid base (297–319, 464–582) into which is inserted a prominent feature denoted the pyramid domain (320–463) (Figs. 1a, 3 and Supplementary Fig. 3). The pyramid domain is specific to Old World arenaviruses and results from residue insertions compared to New World arenaviruses (e.g. MACV) that considerably lengthen the two helices spanning 386–435 (Supplementary Data 1), giving it a characteristic angular shape. At the beginning of the pyramid domain, there is a structural zinc-binding site with coordinating ligands H316, C321, H364, and C366 (Supplementary Fig. 4). Mutational studies using the LASV mini-replicon system showed a general but incomplete reduction in L protein activity upon single-site exchanges to alanine (Supplementary Fig. 4), which further emphasises the structural role of this zinc-binding site. Indeed, sequence comparisons show that this site is specific to the LASV strain Bantou 289 and closely related strains, but not conserved in other LASV lineages or other arenaviruses (Supplementary Data 1). Interestingly, the MACV L protein also contains a zinc-binding site but in a different location, at the pyramid base (Supplementary Fig. 4).

The PB1-like region (Figs. 3a, c, 4e and Supplementary Fig. 3) contains the canonical fingers, palm and thumb with associated conserved polymerase motifs A-F (motif G is 641-RY, motif H is K1237[13]). The catalytic triad of aspartates are D1190 (motif A), D1331 and D1332 (motif C). As previously noted[9], the fingertips (motif F, 1117–1137) are well structured even in the absence of bound promoter (Supplementary Fig. 5), unlike in influenza virus and LACV polymerases[13]. The LASV L PB1-like region is considerably larger than influenza virus PB1 (882 residues compared to 756), mainly due to the so-called Lassa insertion

(830–1069). This includes two flexibly linked modules: (i) a three-helix α-bundle (840–925), which includes an α-ribbon and (ii) the compact pendant domain (943–1042), the latter being only partially visible in our structures (Fig. 3 and Supplementary Fig. 3). The internal connection (887–895) between the α-ribbon and third helix of the α-bundle is disordered as are the flexible linkers before (926–942) and after (1042–1052) the pendant domain (Fig. 3a and Supplementary Fig. 6). These modules were previously observed in the MACV L structure[9], but in different positions and configurations (Supplementary Fig. 6). The PB1-like region has an additional insertion in the fingers, called the finger node (1242–1283), not present in influenza virus PB1, but very similar to the finger node of LACV and likewise involved in binding the 5′ hook (see below) (Fig. 2d)[13]. In all LASV structures, the region 1567–1577 is disordered. Moreover, it is so far unclear whether any protein segment might serve as a priming loop.

The PB2-like region (1593–2217) (Figs. 3a, c, 5c, d and Supplementary Fig. 3) has a similar overall organisation to influenza virus, LACV and Severe fever with thrombocytopenia syndrome bunyavirus (SFTSV) polymerases, with a 'thumb-ring', associated with the core and surrounding the thumb domain. Into the thumb-ring, a 'helical lid' domain (1731–1800) is inserted, which in influenza virus polymerase forces strand-separation during RNA synthesis[22]. This is followed by a short flexible linker to an array of C-terminal domains (1830–2217), visible at lower resolution in the MID-LINK structure (Supplementary Fig. 2), but fully buildable in the well-ordered ELONGATION structure (Figs. 3a–c and 4b–d). This includes the influenza-like split 'mid-link' domain (1831–1903, 2077–2100), into which is inserted the putative cap-binding domain (CBD-like, 1904–2076), followed by a 627-like domain that comprises two helical hairpins (2101–2168) and a terminal, compact β-barrel domain (2169–2208) (Figs. 3a–c, 4c–d and Supplementary Figs. 2, 3). The mid-link and 627-like domains are juxtaposed in the same way in the MID-LINK and ELONGATION structures, suggesting that they are rigidly associated (Supplementary Fig. 2), although possessing some rotational freedom as a whole with respect to the thumb domain (Supplementary Fig. 7). In contrast, the CBD-like domain has considerable rotational freedom, with a difference of ~84° in its orientation with respect to the mid-link domain in the MID-LINK and ELONGATION structures, respectively (Supplementary Fig. 7). The C-terminal region was previously visualised in a dimeric form of MACV L protein (PDB:6KLH), but at an insufficient resolution to build a correct model[9]. The MACV cryo-EM density (EMD-0710) for the C-terminal region is fully compatible with the LASV C-terminal model and shows an extended configuration similar to that observed in the MID-LINK

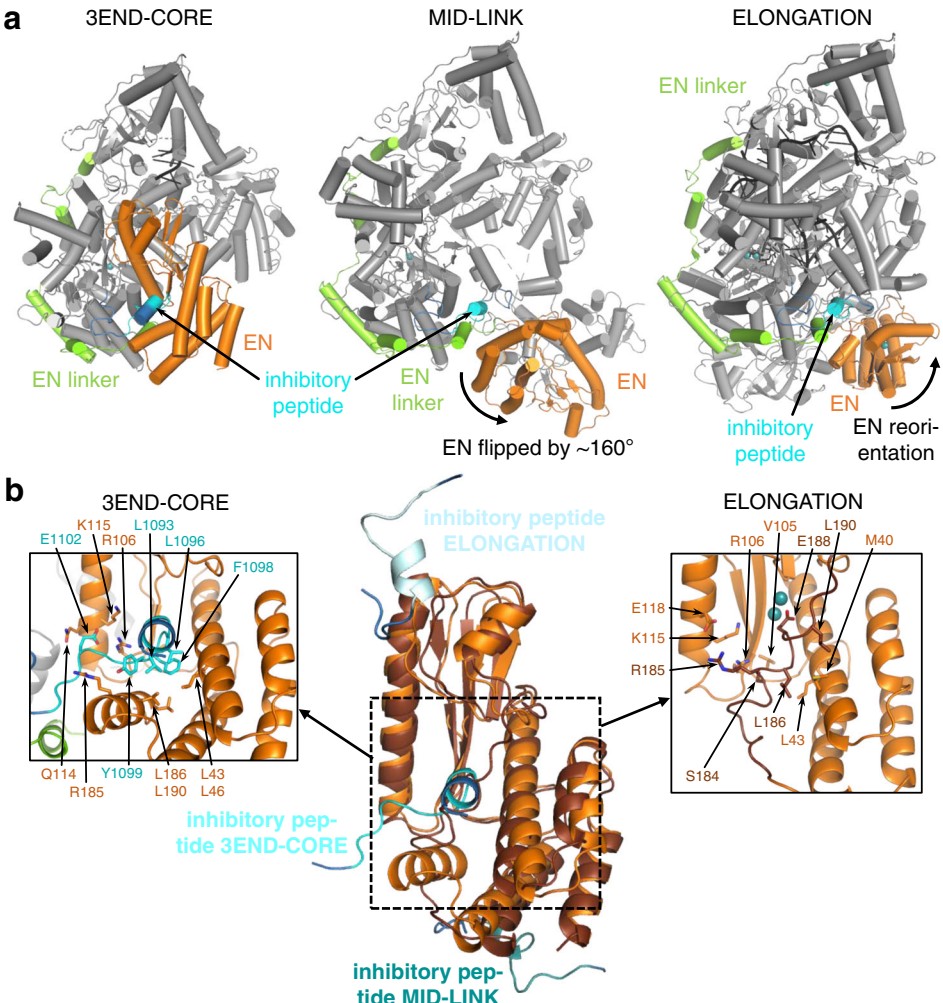

**Fig. 5 Conformational flexibility of the endonuclease domain. a** Overview of the three different conformations of the EN (orange) observed in the 3END-CORE, MID-LINK and ELONGATION structures. The EN linker (green) and the inhibitory peptide (cyan) are highlighted as well. **b** In the middle panel a superimposition of the EN domains of 3END-CORE (orange), corresponding also to the overall conformation of the EN domain in the MID-LINK structure, and ELONGATION (brown) with the respective positions of the inhibitory peptides in teal, cyan and light blue, respectively, is shown. Connections to the fingers domain are indicated in blue. The position of the inhibitory peptide of 3END-CORE is the same as in the APO-ENDO structure, similarly is the same position of the EN observed in both MID-LINK and DISTAL-PROMOTER structures. Right and left panels show close-ups of the autoinhibited EN active sites in the ELONGATION and 3END-CORE structures, respectively. Important residues of the protein-protein interactions are labelled and side chains are shown as sticks. For a focus on the inhibitory loop in the different conformations see Supplementary Fig. 11.

structure (Supplementary Fig. 7). The individual modules of the LASV C-terminal region have similar folds to that of the California Academy of Sciences reptarenavirus (CASV) L protein, despite low sequence homology (Supplementary Figs. 2 and 8)[23]. However, the LASV CBD-like domain is considerably more elaborate, having ~170 residues compared to ~100 residues in CASV (Supplementary Fig. 8). The CASV CBD-like domain is minimalist, comprising a five-stranded mixed β-sheet with a transverse helical hairpin. In LASV, there are significant insertions in the β1-β2 and β3-β4 loops and in the loop of the helical hairpin, that fold together to extend the length of the domain (Fig. 4d and Supplementary Fig. 8). In the ELONGA-TION structure, the CBD-like domain is locked in position by interactions with the EN and core of the polymerase, with the helical hairpin insertion being particularly important for the latter interaction (Supplementary Fig. 9). In this conformation, the canonical cap-binding site between the C-terminal end of strand β1 and the principal transverse helix, as observed in cap-bound CBDs of SFTSV, Rift-Valley fever virus (RVFV) or influenza virus (Supplementary Fig. 8), appears to be partly blocked. Moreover,

the CBD-like and 627-like helical-hairpin domains are particularly poorly conserved across arenavirus L proteins (as opposed to the mid-link and C-terminal β-barrel domain) (Supplementary Data 1 and Supplementary Fig. 8). Indeed, the most conserved region of the CBD-like domain is at the N-terminus of strand β1 (1909-GYAW in LASV). Past and recent mutational analyses of the potential cap-binding aromatic residues of this domain using the LASV mini-replicon system did not identify any transcription-specific residues that might be responsible for cap-binding (Fig. 4d and Supplementary Fig. 8)[24]. Neither have in vitro studies with isolated soluble domains of the CASV and LASV L protein C-terminal region detected any cap-binding activity[23], unlike for phenui- and orthomyxoviruses[25–27]. It, therefore, remains an open question, whether in some, yet to be observed configuration of the L protein, a functional cap-binding site is formed.

**Positional flexibility and regulation of the endonuclease**. We observe three quite different locations of the EN, in each case

differently packed against the core of the L protein (Fig. 5). In the APO-ENDO and 3END-ENDO structures, the EN interacts with the core regions 1137–1142 and 1592–1604 (Supplementary Fig. 10). Mutation to alanine of F1592, which packs on P109 of the EN, resulted in a slight reduction in general L protein activity in the LASV mini-replicon system. A severe, general loss of 90% of activity was observed for L protein mutant P109G (Supplementary Fig. 10). The EN active site itself is exposed to the outside, but access is blocked by residues 1092–1105 from the PB1-like region, which we refer to as the 'inhibitory peptide' (Fig. 5a, b). The arena-conserved 1096-LCFYS motif (Supplementary Data 1) is intimately bound in the EN active site pocket (Fig. 5b and Supplementary Figs. 10, 11) and would thus prevent any substrate RNA binding there (e.g. superposition of LCMV EN structure with bound inhibitor shows overlap[28]) (Supplementary Fig. 11). A similar interaction with the EN is observed in the apo-MACV structure (PDB:6KLD) (Supplementary Fig. 4).

In the MID-LINK and DISTAL-PROMOTER structures, the EN has flipped by ~160° around a hinge between G195-G199. The EN active site faces away from the rest of the polymerase and is exposed to the solvent, free of the inhibitory peptide (Fig. 5a, b). Instead, inhibitory peptide residues 1087–1099, as well as PB2-like segments 1759–1770 (lid), 1852–1860 and 1894–1896 (mid), 2077–2081 and 2089–2091 (link) pack against the back of the EN, stabilising it in its new location. For these two observed positions of the EN, the total buried surface area is comparable, 3005 Å$^2$ (APO-ENDO, autoinhibited) and 2721 Å$^2$ (MID-LINK, free), compatible with there being an equilibrium between the two states as observed in the two different apo-structures.

In the ELONGATION structure, the EN is stabilised in a third position (Fig. 5a, b) with its active site autoinhibited by a completely different mechanism involving the C-terminal region of the EN (173–190). This is redirected so that the 181–188 helix binds in and blocks the EN active site groove, with E188 coordinating, together with E51 and D89[21], two cations in the active site (Fig. 5b and Supplementary Fig. 10). The inhibitory peptide remains at the same place with respect to the polymerase core, but due to the reorientation of the EN it packs against a different site on the EN, with, for instance, K1094 making a salt-bridge with E70 and Y1099 stacking against P81 (Fig. 5b and Supplementary Fig. 11). Diverse other regions of the L protein also contact the EN (Fig. 3b and Supplementary Fig. 9) and the total buried surface area of the EN in this location is 4060 Å$^2$.

To investigate the function of the inhibitory peptide, we performed a mutational analysis of residues 1092–1105 as well as interacting residues of the EN domain, as observed in the APO-ENDO and 3END-CORE structures (Supplementary Figs. 10, 11). We observed a severe general defect in L protein function for a number of mutants both in the EN domain (L43G/N, L46G/N, V105G, R106K, K115A, R185A, L186G, L190G/N) and in the 'inhibitory peptide' (L1093S, L1096A/N, C1097G, F1098A/S, Y1099A, E1102A) (Supplementary Fig. 11). For hantavirus L protein it was shown that an active EN can lead to RNA degradation and therefore lower protein expression levels[29]. To exclude that the general defect of the mutants of the 'inhibitory peptide' interaction is caused by RNA degradation due to elevated activity of LASV EN, we combined the previous mutations with the EN inactivating mutation D89A[30], without observing any change in phenotype (Supplementary Fig. 12). Additionally, using in vitro polymerase assays only residual polymerase activity was detectable for L protein mutant E1102A and no activity for mutant Y1099A (Supplementary Fig. 13). We conclude that the 'inhibitory peptide' and other tested residues involved in the interaction play a general role in L protein activity but are not selectively important for transcription, this being consistent with the diversity of interactions we see for these residues when

comparing all observed conformations of the L protein (Fig. 5). Comparing the apo- and pre-initiation structures suggests that either promoter binding or mutations in the inhibitory peptide release the EN from autoinhibition. We tested this hypothesis by assaying purified L proteins with mutations in the inhibitory peptide for EN activity in vitro, using capped or uncapped RNA substrates and with either no promoter, 3′ end only, 5′ end only or both promoter RNAs present (Supplementary Fig. 14). The EN active site mutant E102A and the addition of the nuclease inhibitor DPBA served as negative controls. We found that, for the wild-type L protein, the only situation where weak EN activity is reproducibly detectable is when the 5′ end only or both promoter ends are bound, and the same is true for the L protein with mutations Q114A or E1102A, which are probably not sufficient to disrupt inhibitory peptide binding.

In summary, in both the apo- and early elongation states, the EN is autoinhibited, but by different mechanisms involving binding in the active site of either the 'inhibitory peptide' 1092–1105 or the C-terminal helix of the EN, respectively. Whilst 5′ end or full promoter binding partly activates the EN, consistent with the structure and presumed functional role in cap-dependent transcription priming of the pre-initiation state, its low intrinsic activity in vitro under any conditions tested by us and others[20,21,31] suggests that the mechanism of EN activation may be more complex than expected from the currently available structural data.

**3′ end binding in the secondary site**. Incubation of LASV L protein with vRNA 3′ end nucleotides 1–16 yielded the currently highest resolution structure (3END-CORE, 2.70 Å). It features specific binding of nucleotides 1–7 in a tunnel under the pyramid (Fig. 1b, c). This site corresponds to the secondary 3′ end site previously observed for influenza virus polymerase[16,17], LACV L[13] and MACV L[9]. The excellent cryo-EM density enables unambiguous base identification of nucleotides 2–5 (CGUG) of the 3′ end and placing of several water molecules in the protein–RNA interface (Supplementary Fig. 15). However, G1 and nucleotides 6–7 (UC) have poor density. In the MACV L structure, G1 is better defined perhaps due to its stabilisation by stacking on protein residue W534, which is substituted by L540 in LASV L (Supplementary Data 1). Nucleotides 3–5 of the 3′ end form a particularly compact arrangement with a direct interaction between G3 O6 and G5 N2, and U4 stacking underneath (Fig. 1c and Supplementary Fig. 15).

Several specific protein–RNA interactions are made with conserved arenavirus residues such as K332, D380, L502 and K509 from the pyramid and Y1450, R1452 and S1626 from the thumb and thumb-ring domains, thus involving the PA-, PB1 and PB2-like regions, as in influenza virus and LACV (Fig. 1c). Mutational analysis of the residues interacting with the 3′ end in the secondary binding site using the LASV mini-replicon system revealed a general defect in L protein function upon introduction of double mutations N331A/K332A and Y1450A/R1452A, whereas mutations L502A, K509A and R1622A did not interfere with L protein activity (Fig. 1d). Purified L protein mutant Y1450A/R1452A exhibited significantly reduced 3′ end binding ability compared to wild-type L protein (Fig. 1e). However, this mutant maintained polymerase activity in the presence of the 19 nt 3′ and 20 nt 5′ promoter RNAs and, in contrast to the wild-type L protein, showed polymerase activity with only the 19 nt 3′ promoter RNA present (Supplementary Fig. 16). This strongly suggests that in the wild-type L protein, in the absence of the 5′ end, the 3′ end is tightly sequestered in the secondary binding site and does not enter the active site (see discussion). In the presence of a 47 nt hairpin RNA containing the connected 3′ and 5′

promoter sequences of LASV, L-Y1450A/R1452A showed significantly reduced polymerase activity (Supplementary Fig. 16). This shows that 3′ end binding in the secondary site is required for efficient RNA synthesis, either to sequester the template 3′ end after passing through the active site and/or to prevent the unbound 3′ end from forming double-stranded RNA with the template 5′ end or the product RNA.

**Full promoter binding**. The PRE-INITIATION structure shows the full promoter (5′ nts 0–19, 3′ nts 1–19) bound to the LASV L protein and reveals several significant conformational changes that occur upon promoter binding (Fig. 2 and Supplementary Fig. 1). The LASV vRNA 5′ and 3′ ends are highly complementary over 19 nucleotides with only two mismatches at positions 6 and 8 (Supplementary Table 1). In addition, the 5′ end carries an extra nucleotide (G0) arising from a prime-and-realign mechanism during replication initiation[4,5,32–34]. As expected, when bound to the L protein, the promoter does not adopt a fully double-stranded conformation but forms a structure resembling that observed for influenza virus and LACV (Fig. 2a, c, d, e and Supplementary Fig. 17). Nucleotides 12–19 from both strands form a distal 8-mer canonical A-form duplex, whereas nucleotides 1–11 of each end are single-stranded, which is consistent with previous mutational studies[35]. Nucleotides 0–9 of the 5′ end form a compact hook structure linked to the duplex region by nucleotides 10–11. The internal secondary structure of the hook differs from that of influenza virus and LACV polymerases by only having one canonical base-pair (C1-G7) upon which G2 and C3 are consecutively stacked on one side and G8 and G9 on the other side (Fig. 2d and Supplementary Fig. 17). The loop of the hook comprises nucleotides 4–6, with A4 stacking on C6. The hook makes extensive interactions with conserved residues from numerous different loops from both the PA-like and PB1-like regions of the L protein (e.g. residues 470–474, 518–526, 860–861 and 1255–1265) (Fig. 2d and Supplementary Movie 1). Several of these loops only become structured upon promoter binding. Of note, three aromatic residues are involved, Y471 (conserved in Old World arenaviruses only), Y526 (Y or F in all arenaviruses) and Y574 (conserved in all arenaviruses) (Supplementary Data 1). Y471 interacts with the phosphate of G10 and Y574 with the base of G9. Y526 extends and stabilises the central stacked backbone of the hook by packing on G9, with I665 playing a similar role on the other end by stacking against C3. G0 is base-specifically recognised by the carbonyl-oxygen of S470 and is sandwiched between P297 and R473, which could potentially interact with the terminal triphosphate (not present in the 5′ RNA used for this structure) (Supplementary Fig. 17). Mutational analysis of the 5′ binding site using the LASV mini-replicon system reveals that most mutations cause a severe general defect in L protein activity (Supplementary Fig. 1). Complete loss of function was observed for mutants Y474A, V514G/K515A, R525A/Y526A and K681A. An activity reduction to ~20% was observed for mutants R473A/T474A, Q551A/K552A, Y574A and K1263A/T1265A. These results indicate that 5′ end binding is important for the general function of LASV L protein during both viral transcription and genome replication.

The 3′ strand of the distal duplex specifically interacts with two regions of the L-protein pyramid (Figs. 2b and 6a). One is the 340-loop, which orders upon binding, with the arenavirus-conserved motif 340-NTRR making several major groove contacts with bases and phosphates of the 3′ strand nucleotides 12–15 (and R337 with phosphate of C10). The second interacting region involves the Old World arenavirus-specific extended helices at the top of the pyramid, including W395, R399 (conserved in all Old World arenaviruses) and K423 contacting

the backbone phosphates of 3′ strand nucleotides 16–17 (Fig. 2b and Supplementary Movie 1). The single-stranded 3′ end nucleotides 11 to 7 are directed towards the polymerase active site, but 1–6 are not visible (Fig. 2a, c, e). Arenavirus-conserved R1561 interacts base-specifically with C9, which also stacks on chemically conserved R1622, both residues being from the thumb domain (Fig. 2b and Supplementary Movie 1).

Apart from the induced fit ordering of several promoter-interacting loops, there are also more global rearrangements. Firstly, distal duplex binding causes a major rotation of the entire pyramid by ~21.6°, enabling the summit helices to contact the 3′ strand as described above (Fig. 6a). This rotation does not occur when just the 3′ end is bound in the secondary site. Secondly, the α-bundle rotates slightly (4.5°, PRE-INITIATION versus APO-RIBBON) to allow interaction of the 860-region (860–861) with the 5′ hook (Fig. 6a). Thirdly, the pendant domain (943–1040) becomes stably positioned by packing against the rotated pyramid domain and the thumb and thumb-ring (Fig. 6b). The pendant domain helix spanning residues 953–965, runs parallel to 3′ nucleotides 9–12, but with only two direct interactions. R957 contacts the phosphate of A12 contributing to stabilisation of the 3′ end interacting 340-loop and K960 is close to the phosphate of U11 (Fig. 2b). The pendant domain was first visualised in the MACV apo-L structure in a very different position that would superpose with the promoter duplex region as well as the α-bundle, as observed in the promoter-bound LASV L structure (Supplementary Fig. 6). Similarly, the α-bundle of MACV apo-L (with its different topology, see Supplementary Fig. 6) superposes with the pendant domain in LASV L, showing that these flexible and linked domains must rearrange from the apo-state observed in MACV L. Finally, the position of the pendant domain in the LASV PRE-INITIATION structure is incompatible, due to significant steric overlap, with the location of the inhibited EN in the APO-ENDO or 3END-ENDO structures (Fig. 6c). This important observation provides a plausible rationale for full promoter binding (as opposed to just the 3′ end binding in the secondary site) inducing a flip of the EN to the alternative uninhibited location observed in the MID-LINK and DISTAL-PROMOTER structures (see above).

**Elongation structure**. To determine a structure of functionally active LASV polymerase in early elongation, we incubated promoter-bound L protein with a 10-mer uncapped primer, 5′-AAUAAUACGC-3′ together with ATP, GTP, CTP and non-hydrolysable UMPNPP (denoted $U_{PNPP}$). Biochemical analysis (Supplementary Fig. 18) shows that various products are formed depending on whether (i) the 3′ terminal triplet of the primer hybridises with the 3′ end of the template (3′-GCGUGUCA…) giving 14-mer 5′-AAUAAUA<u>CGC</u>*CACAG*U$_{PNPP}$ or 18-mer: 5′-AAUAAUA<u>CGC</u>*CACAG[U]GGA*U$_{PNPP}$ products or (ii) just the 3′ terminal nucleotide hybridises, giving 16-mer 5′-AAUAAUAC G<u>C</u>*GCACAG*U$_{PNPP}$ or 20-mer 5′-AAUAAUACG<u>C</u>*GCACAG[U] GGA*U$_{PNPP}$ products (here, underline indicates primer hybridisation, italics indicate incorporated nucleotides, [U] misincorporation at an A in the template) (Supplementary Fig. 18). In each case, the longer product is formed if the A8 in the template is read-through by misincorporation. The most prominent products are the 14-mer and 18-mer. Upon plunge freezing on EM grids and performing 3D single-particle reconstruction, the sample gave a major class showing the stalled (i.e. pre-incorporation), early elongation state at 2.92 Å resolution. The high quality of the density allows unambiguous assignment of the template and product sequences. As expected, incoming UMPNPP is observed at the +1 position base-pairing with A8 of the template. There is good density for seven bases of the product

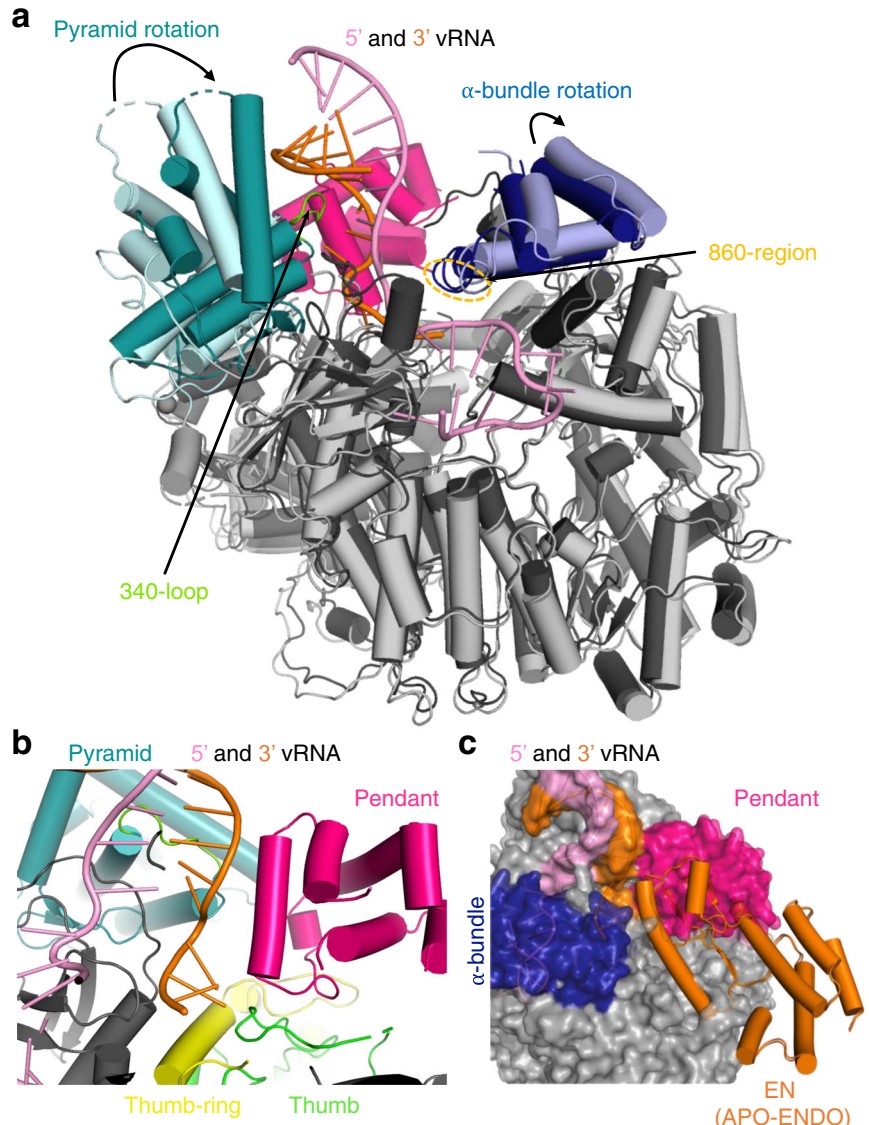

**Fig. 6 Global rearrangements upon promoter binding. a** PRE-INITIATION (dark grey, teal, dark blue, hot pink) and APO-RIBBON (light grey, light cyan, light blue) structures are superimposed. The pyramid and α-bundle rotations between apo- and promoter-bound structures are indicated as well as the 340-loop and the 860-region. Promoter RNA is shown in pink and orange. **b** Close-up of the interaction site between promoter RNA (pink and orange) and the pendant (hot pink), thumb-ring (yellow) and thumb (green) domains in the PRE-INITIATION structure. The pyramid domain is shown in teal. **c** Superposition of the PRE-INITIATION and APO-ENDO structures. PRE-INITIATION is presented as a transparent surface in grey with the pendant (hot pink) and α-bundle (blue) as well as the 5′ (pink) and 3′ (orange) vRNA highlighted in colour. The EN domain of the APO-ENDO structure is shown as an orange ribbon, which overlaps with the pendant domain volume of the PRE-INITIATION structure.

(5′-….N$_1$N$_2$CGCACAG), with the preceding two, which are unpaired with the template, having poorer density, prohibiting unambiguous identification (Supplementary Fig. 19). However, since the map density corresponding to N$_2$ looks more like an A, the product is most likely the 14-mer. Products that would contain mismatches due to read-through, maybe less stably bound to the polymerase. The active site cavity contains an 8-mer duplex from position +1 to −7 (+1 corresponding to the UMPNPP) (Fig. 4a, e, f), whereas as observed for related viral polymerases[14,22] a 10-mer duplex (positions +1 to −9) is expected to fill the active site cavity before strand-separation. Nucleotides 1–11 of the template are visible, corresponding to positions +4 to −7. The duplex region of the promoter has melted due to translocation of the template but nucleotides 0–11 of the 5′ end remain bound in the hook confirmation as described

above. Consistent with the distal duplex being absent, the pyramid is not rotated and a second template 3′ end (nucleotides 1–7) is actually bound in the secondary site (Fig. 4b). Whilst this is likely an artifact of performing the RNA synthesis reaction with an excess of template, it does confirm that secondary 3′ end binding is compatible with elongation and consistent with the template docking in this site after exiting the active site cavity, as observed for influenza virus polymerase[18]. The pendant domain is not visible and the α-bundle only has weak density, probably due to the absence of the distal promoter duplex.

The configuration of the polymerase active site, as well as the binding of the incoming nucleotide and template, are canonical, involving conserved motifs A–D and the fingertips (motif F) (Fig. 4e and Supplementary Movie 2). The triphosphate and terminal 3′ OH of the product strand are coordinated by two

manganese ions (A and B), held in place by D1190 (motif A) and 1331-DD (motif C). K1373 of motif D also contacts the γ-phosphate. Motif F residues R1131, positioned by Q1294 (motif B), and L1133 stack under the incoming nucleotide template bases at the +1 position, respectively, while K1124 (motif F) contacts the O4 of the incoming nucleotide (Supplementary Movie 2).

Compared to the unoccupied active site in the PRE-INITIATION structure, only minor adjustments to the active site loops occur, the most significant being the displacement of the central β-strands of the fingertips loop by about 2 Å to make room for the +1 base-pair to stack on R1131 and L1133 (Supplementary Fig. 20). Unlike in influenza virus polymerase[22], motif B does not change conformation between the occupied and unoccupied states of the active site. However, to accommodate the growing template-product duplex, the helical lid (1731–1805) has to be displaced out of the active site cavity by about 8 Å (Supplementary Fig. 20). Coupled with the lid movement, the sharply kinked (requiring conserved G1595) pair of consecutive helices α52-α53 (1579–1611) also translates in the same direction, with helix α53 forming one side of the active site cavity close to the distal part of the product strand. Interestingly, the first visible base of the product (position −9) is packed against T1583 from α52, which could therefore play a role in strand-separation rather than the helical lid itself (Supplementary Fig. 20).

More generally, the conformation of the active elongating polymerase is stabilised by a number of new interactions between distant regions of the L protein sequence. For instance, in the new position of the helical lid, residues 1764–1766 contact the EN at F85 (close to the EN active site), contributing to the interactions which stabilise the EN in its third location (see above). Residues 811–820, disordered in all other structures, interact with EN linker 195–199, again only possible with the EN in its new location. Similarly, the inhibitory peptide residues 1087–1091 change conformation to allow simultaneous interaction with the EN (1089-TT with D129 and S82) and with the kink between α52-α53 (A1091 with T1591) (Fig. 5a and Supplementary Fig. 11). Most dramatic, is the stabilisation of the entire C-terminal domain, which, together with the exposed end of the palm, forms a ring with a ~30 Å diameter central pore, a putative product exit channel (Fig. 4b). The EN buttresses the proximal part of the CBD-like domain (e.g. Q32 with A1911) as well as the mid-link domain (e.g. A171 with K1895 and E34 with K1891), whereas the distal part of the CBD-like domain interacts with numerous loops from the polymerase core including EN linker residues 230–232 (e.g. H232-Q2045), fingers domain residues 793–799, 802–805 (e.g. V802-Y2030), 1215–1216 (e.g. D1216-K2062, K1215- E2053) and palm domain residues 1314–1318 (e.g. Y1314-Q2045) (Supplementary Fig. 9). The total buried surface area between the polymerase core and the CBD-like domain is 2884 Å$^2$. The extreme C-terminal 627-like domain (mainly the β-barrel and to a lesser extent, the helical hairpins) make interactions with multiple regions, notably 1715–1722 and 1812–1816 of the thumb-ring (e.g. F1715-Y2176/V2145, F1716-V2189, D1722-S2191/S2192, R1816-D2143, S1812/L1815-G2175), residues 691–694 of the helical region (e.g. M691-G2193) and the palm domain 1390–1392 loop (e.g. W1390-R2197) (Supplementary Fig. 9). The total buried surface area of the β-barrel domain is 1472 Å$^2$. Interestingly, W2170, R2197 and R2201, whose mutation leads to a transcription-specific defect[24], are intimately involved in the interface together with W1390 in the 1390-loop. Even though mutation of W1390 to alanine in a previous study did not impair overall L protein activity[36], from the structure we would not expect a small hydrophobic alanine residue to disturb the remaining contacts between these domains. Additionally, residues G1391 and D1392 were shown to be selectively

important for viral transcription[36], further emphasising the importance of this interaction site. Residue Y2176, also identified as being selectively important for viral transcription[24] interacts with the thumb-ring residue F1715 (Supplementary Fig. 9). These data suggest that the configuration of the C-terminal region and its interaction with the core, as observed in the ELONGATION structure is critical for transcription.

**Discussion**

Previous biochemical studies on LASV L[7] and MACV L[8,37] proteins have revealed certain features of promoter binding to arenavirus polymerases and the impact on RNA synthesis activity. In RNA binding experiments, it was shown that MACV L makes a tight complex with the 3′ promoter strand with the identity of nucleotides 2–5 being particularly important[8]. This corresponds exactly with the binding specificity of the 3′ end secondary site seen in our structural analysis. For both MACV and LASV, the most efficient in vitro RNA synthesis activity was observed using both 19-mer strands in 1:1 ratio as in the native promoter[7,37]. For MACV L protein weak activity, which could be enhanced with a GpC primer, was also observed in presence of only the 3′ strand[8]. The more specific requirements found for optimal unprimed RNA synthesis by LASV L were (i) the presence of the terminal non-templated G0 base on the 5′ strand (i.e. 0–19); (ii) the two mismatches at positions 6 and 8 in the S segment promoter (only one in the L segment), rather than a perfect proximal duplex; (iii) a sufficiently long distal duplex region, preferably the full 19-mer[7]. For MACV, it was further shown that the G0 phosphates were not essential and that enhancement was achieved with 3′ truncated 5′ ends (e.g. maximal activity for 0–12 mer), although these experiments differed in that a GpC primer was systematically used[37]. These observations are consistent with our structural analysis as well as the notion that the default mode of binding of the 3′ end alone is in the secondary site and that full-promoter binding or a primer is required to dislodge it and permit RNA synthesis. More recently, it has been confirmed for both LASV and MACV L that de novo RNA synthesis is indeed enhanced by the presence of the 5′ end, but, surprisingly, it was reported that cap-dependent transcription was inhibited by the 5′ end[9]. This raises the question of the exact role of the 5′ end binding in arenavirus L proteins.

In LACV and influenza virus, 5′ end hook binding is required to order the fingertips loop into a functional configuration in the polymerase active site[13,14]. This does not appear to be the case for arenavirus L proteins[9] (Supplementary Fig. 5). In addition, for the influenza virus, it has been shown that 5′ end binding stimulates EN activity, probably by favouring the transcription active configuration of the polymerase over the replicase conformation[19]. Evidence given above that 5′ end (or full promoter) binding stimulates EN activity suggests that a similar conformational change mechanism may operate for arenavirus L proteins. Finally, whereas 5′ end binding is required in orthomyxoviruses for poly(A) tail generation during transcription[18], arenavirus L proteins terminate transcription by a very different mechanism without poly(A) tail synthesis[38–41]. To investigate the functional consequences of 5′ end hook binding further, we performed polymerase activity assays either (i) with the 3' promoter end alone, (ii) the 3′ end together with the 5′ (nts 0–19) end or (iii) the 3′ end together with a truncated 5′ (nts 0–12) end. In each case, the assays were performed in the presence or absence of 3 nt or 10 nt long uncapped primers (Supplementary Fig. 21). We used uncapped primers as we could not detect any difference in primed product formation between uncapped (hydroxylated or tri-phosphorylated 5′) and cap0-capped primers for LASV L (Supplementary Fig. 22). Under the conditions of the reaction, no

products were formed by the 3′ end alone unless a primer was present (Supplementary Fig. 21). Adding the promoter 5′ end (nts 0–19) led to strong product formation even in the absence of primer. When the 5′ end was truncated (nts 0–12), unprimed product formation was significantly reduced, whereas primed product formation was comparable to the corresponding conditions with the full 5′ (nts 0–19) RNA. In the absence of the 5′ end, structural and biochemical data show that the 3′ end preferentially binds tightly in the secondary binding site. Bearing this in mind, we interpret our activity results to show that 5′ end binding and/or the presence of a primer (capped or uncapped), that can hybridise with and stabilise the 3′ end in the polymerase active site, stimulates RNA synthesis activity, presumably by promoting 3′ end binding in the active site rather than the secondary site. Indeed, the major rotation of the pyramid domain induced by distal duplex binding shears the two sides of the 3′ end-binding groove and prevents closure around the RNA, thus disfavouring the secondary site 3′ end binding when the full promoter is bound. Since the promoter duplex no longer exists during elongation, this mechanism does not prevent the template 3′ end rebinding in the secondary site after going through the active site as observed for influenza polymerase[18].

In conclusion, our structural and functional results support the hypothesis that full-promoter binding, including the 5′ hook and distal duplex, induces the functionally ready pre-initiation state via the conformational changes described above, at the same time as releasing the EN from autoinhibition (Fig. 7). Given that our activity results are independent of whether the primer is capped or uncapped (Supplementary Fig. 22), we do not think they shed light on true cap-dependent transcription per se. It remains unclear, whether the assays presented by Peng et al.[9] truly reflect cap-dependent transcription, as the respective control (i.e. non-capped primer) was not included. Indeed, we have not been able to recapitulate true cap-dependent transcription with the LASV polymerase, due to its weak or non-existent endonuclease and cap-binding activities, emphasising that the mechanism of cap-snatching and cap-dependent transcription for arenavirus L protein remains enigmatic. This is even more remarkable considering the shortness of the capped primers used[3], which it is difficult to imagine being bound in a canonical way by the CBD-like domain as well as reaching into the active site to hybridise with the template, a situation reminiscent of the Thogoto virus polymerase[42].

The autoinhibited EN conformations appear in L protein configurations where the EN activity is not expected to be required and indeed could potentially be detrimental (Fig. 7). On the other hand, in the promoter-bound pre-initiation state, EN activity is required[30,31], presumably to generate capped primers. Correspondingly, in this state, we observe that the EN is not autoinhibited, although how it might act in collaboration with the putative CBD is unclear. Whilst these biochemical experiments support the structure-based hypothesis that 5′ end only or full promoter binding activates the EN due to its displacement by the pendant and α-bundle domains, the observed EN activity is very weak (Supplementary Fig. 14), consistent with the barely detectable in vitro activity of the isolated EN domain[20,21,31]. This suggests that some other L protein configuration or possibly a host factor may be required to fully activate the EN inside infected cells. Similarly, interaction with a currently unknown host cap-binding protein may be required to

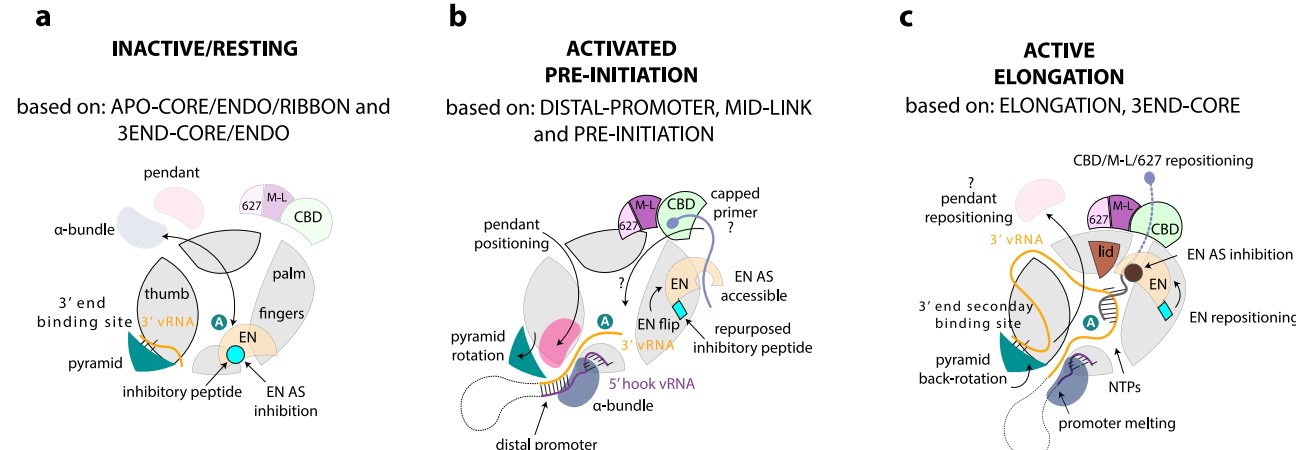

**Fig. 7 Schematic diagram of conformational changes in Lassa virus L protein associated with promoter binding and RNA synthesis activity. a** On the surface of the inactive/resting L protein core, there is mutually exclusive positioning (black double-arrow) of either the α-bundle and pendant (APO-RIBBON structure) or the EN domain (APO-ENDO, 3END-ENDO structures). When placed on the core, the inhibitory peptide (cyan) autoinhibits the EN domain by binding in its active site. In the absence of the 5′ vRNA, the 3′ vRNA binds preferentially, base-specifically, into a distinct secondary 3′ RNA binding site between the pyramid and thumb domains (3END-ENDO/CORE structures). **b** Upon full promoter binding, major conformational changes occur (based on PRE-INITIATION, DISTAL-PROMOTER, MID-LINK structures). The 5′ end nucleotides 0–9 are bound in a hook-like conformation in a specific pocket outside the active site. The distal promoter (formed by 3′ and 5′ vRNA nucleotides 12–19) is positioned by tight association with the α-bundle and pendant, with concomitant rotation of the pyramid domain. This forces release of the 3′ vRNA from the secondary binding site, allowing it to be directed towards the RNA synthesis active site (marked by the white A in the teal circle). Positioning of the pendant domain next to the distal promoter displaces the EN domain, which relocates to the other end of the L protein core with the inhibitory peptide contacting its surface leaving the EN active site accessible. In this configuration, the EN is presumed to be in close vicinity to the CBD-like domain and could potentially cleave an incoming capped RNA to generate a transcription primer by 'cap-snatching'. How the capped primer associates with the CBD-like domain and how it is navigated towards the active site to initiate transcription remains elusive. **c** Upon transition to the elongation state, the distal promoter duplex melts and the pendant domain is released, which allows the pyramid to rotate back and re-establish the availability of the secondary 3′ end binding site (ELONGATION structure). We presume the 3′ vRNA template, after exiting the active site, wraps around the L protein core and rebinds to the secondary 3′ end binding site, as described for influenza virus polymerase complex[18]. The EN repositions once again and together with the mid-link and CBD-like domains form a highly structured ring around the putative product exit channel. The EN active site is again autoinhibited, this time by the relocation of its C-terminal helix (181–188) to the active site.

present capped RNAs to the L protein. Identification of how arenavirus L proteins access host capped RNAs is a key requirement for further understanding of the mechanism of transcription.

Whilst this paper was under review, two papers were published describing the cryo-EM structures of the complex between the L protein and the multifunctional Z proteins of MACV and LASV[43] and of Junin virus[44]. The Z protein is known to inhibit RNA synthesis[7,45,46]. All L-Z structures show that the central, folded region of the Z protein, which includes the RING domain, binds to the external rim of the palm domain of L. In each case, the L protein is in an inactive state, with at most the 3′ vRNA end bound in the secondary site. Only the polymerase core is visible, except in a dimeric form of the MACV L-Z complex, where the C-terminal domain, including the CBD-like domain, is ordered. Using modelling of the RNA synthesis active state of the L protein and hydrogen-deuterium exchange experiments, Xu et al. conclude[43] that the mechanism of RNA synthesis inhibition is allosteric, with Z protein binding conformationally restraining two key polymerase active site motifs. On the other hand, Kang et al. suggest[44] that Z protein binding to the Junin virus L protein 680-loop (690-loop in LASV L) locks the L protein in an inactive state with Z positioned to block the product exit channel. A key contact mediating the LASV L (strain G3278-SLE-2013) and Z interaction involves Z protein/W35 contacting L protein/M694-F1381-W1392 with similar interactions in MACV and Junin virus complexes (MACV: Z protein/W43 with L protein/F689-F1378-M1389; Junin virus: Z protein/W43 with L protein/F688-F1377-M1388). We note that the W1390 in our LASV strain Bantou 289 (W1392 in strain G3278-SLE-2013) is also central to the interaction between the 1390-loop and the extreme C-terminal 627-like domain (R2197) as observed in our early elongation state structure (see above and Supplementary Fig. 23). Indeed, superimposition of the active ELONGATION structure with the L-Z complex structure clearly reveals that the binding sites on the L protein core region (involving the 690-loop and the 1390-loop) of the 627-like domain and the Z protein, respectively, substantially overlap, showing that this binding is mutually exclusive (Supplementary Fig. 23). Furthermore, we see no significant change in conformation in the 690-loop between the two structures, and the Z protein, although not resolved in full length in the L-Z complex structures, does not appear to block the putative product exit channel (Supplementary Fig. 23). In addition, binding of Z protein or the 627-like domain of L to the palm domain might reduce the conformational flexibility of the active site motifs, without this necessarily causing inhibition of RNA synthesis. We, therefore, suggest an alternative explanation for the mechanism of inhibition of L by Z: binding of Z protein to L protein sterically prevents the establishment of the RNA synthesis active state of L, with its well-defined ring of peripheral domains surrounding the exit channel. These observations emphasise the importance of determining structures of active states of the L protein in order to understand functional mechanisms.

## Methods

**Expression and purification of LASV L protein.** The L gene of LASV Bantou 289 (accession no. MK044799) contains a StrepII-tag at an internal position (after residue 407, 407strep) or a StrepII-His tandem tag at the C terminus (Cstrep) was cloned into a pFastBacHT B vector[7]. If indicated, mutations were introduced by mutagenic PCR before cloning. Using DH10EMBacY E. coli cells[47,48], recombinant baculoviruses were produced and subsequently used for protein expression in Hi5 insect cells[7]. The harvested Hi5 insect cells were resuspended in Buffer A (50 mM HEPES(NaOH) pH 7.0, 1 M NaCl, 10% (w/v) Glycerol and 2 mM dithiothreitol), supplemented with 0.05% (v/v) Tween20 and protease inhibitors (Roche, cOmplete mini), lysed by sonication and centrifuged two times (20,000 × g for 30 min at 4 °C). Soluble protein was loaded on Strep-TactinXT beads (IBA) and eluted with 50 mM Biotin (Applichem) in Buffer B (50 mM HEPES(NaOH) pH 7.0, 500 mM NaCl, 10% (w/v) Glycerol and 2 mM dithiothreitol). L protein-containing fractions

were pooled and diluted 1:1 with buffer C (20 mM HEPES(NaOH) pH 7.0) before loading on a heparin column (HiTrap Heparin HP, GE Healthcare). Proteins were eluted with Buffer A and concentrated using centrifugal filter units (Amicon Ultra, 30 kDa MWCO). The proteins were subsequently used for biochemical assays and structural studies. For endonuclease assays, the L proteins were further purified by size-exclusion chromatography (Superose 6, GE Healthcare) in buffer B. Pure L proteins were concentrated as described above, flash frozen and stored at −80 °C.

**In vitro LASV L complex reconstitution for cryo-EM**
*APO-structures.* The LASV L-Cstrep protein was first injected onto a Superdex 200 Increase 3.2/300 column (GE Healthcare) equilibrated in 40 mM HEPES pH 7.4 (4 °C), 500 mM NaCl, 10 mM MgCl₂ and 1 mM TCEP. About 50 µl fractions were collected and the protein was eluted at a 2 µM concentration. Protein was diluted to 0.7 µM and aliquots of 3 µl were applied to Quantifoil R1.2/1.3 Au 300 mesh grids, immediately blotted for 2 s and plunged into liquid ethane using an FEI Vitrobot IV (4 °C, 100% humidity).

*PROMOTER-DUPLEX and MID-LINK structures.* The LASV L-Cstrep protein was first injected onto a Superose 6 Increase 3.2/300 column (GE Healthcare) equilibrated at 4 °C in 40 mM HEPES pH 7.4, 400 mM NaCl, 10 mM MgCl₂ and 1 mM TCEP. About 50 µl fractions were collected and the protein was eluted at a 2 µM concentration. Protein was diluted to ~0.9 µM and mixed with a 1.3-fold molar excess of truncated promoter vRNAs (5′ nts 10–19, 3′ nts 1–19) (Supplementary Table 1) for 10 min at 4 °C. Aliquots of 3 µl were applied to Quantifoil R2/2 Au 300 mesh grids, immediately blotted for 2 s and plunge frozen into liquid ethane using an FEI Vitrobot IV (4 °C, 100% humidity).

*3′END structures.* The LASV L-Cstrep protein was first injected onto a Superose 6 Increase 3.2/300 column (GE Healthcare) equilibrated at 4 °C in 40 mM HEPES pH 7.4, 250 mM NaCl, 10 mM MgCl₂ and 1 mM TCEP. About 50 µl fractions were collected and the protein was eluted at a 2 µM concentration. Protein was diluted to ~1.8 µM and mixed with threefold molar excess of 3′ (1–16) vRNA (Supplementary Table 1) for 10 min at 4 °C. Aliquots of 3 µl were applied to Quantifoil R2/2 Au 300 mesh grids, immediately blotted for 2 s and plunge frozen into liquid ethane using an FEI Vitrobot IV (4 °C, 100% humidity).

*PRE-INITIATION structure.* The LASV L-Cstrep protein with a concentration of 1.4 µM in assay buffer (100 mM HEPES(NaOH) pH 7.0, 50 mM NaCl, 50 mM KCl, 2 mM MnCl₂ and 2 mM dithiothreitol) was mixed sequentially with single-stranded 5′ (0–19) vRNA and single-stranded 3′ (1–19) vRNA in 1.2-fold and primer St1 in 7.1-fold molar excess (all RNAs are listed in Supplementary Table 1). After 45 min incubation on ice, the reaction was started by the addition of NTPs (0.25 mM GTP/ATP). After incubation at 30 °C for 2 h, 3 µL of the reaction was applied to glow-discharged Quantifoil R2/1 Au G200F4 grids, immediately blotted for 2 s using an FEI Vitrobot Mk IV (4 °C, 100% humidity, blotting force–10) and plunge frozen in liquid ethane/propane cooled to liquid nitrogen temperature.

*ELONGATION structure.* The LASV L-Cstrep protein with a concentration of 3 µM in assay buffer (100 mM HEPES(NaOH) pH 7.0, 50 mM NaCl, 50 mM KCl, 2 mM MnCl₂ and 2 mM dithiothreitol) was mixed sequentially with single-stranded 5′ (0–19) vRNA and single-stranded 3′ (1–19) vRNA in 1.7-fold and primer C8 in 3.3-fold molar excess (all RNAs are listed in Supplementary Table 1). After 45 min incubation on ice, the reaction was started by addition of NTPs (0.25 mM GTP/ATP/UMPNPP and 0.125 mM CTP). After incubation at 30 °C for 2 h, 3 µL of the reaction was applied to glow-discharged Quantifoil R2/1 Au G200F4 grids, immediately blotted for 2 s using an FEI Vitrobot Mk IV (4 °C, 100% humidity, blotting force–10) and plunge frozen in liquid ethane/propane cooled to liquid nitrogen temperature.

**Electron microscopy**
*APO-, DISTAL-PROMOTER, MID-LINK, PRE-INITIATION and ELONGATION structures.* The grids were loaded into an FEI Tecnai Krios electron microscope at the Centre for Structural Systems Biology (CSSB) Cryo-EM facility, operated at an accelerating voltage of 300 kV and equipped with K3 direct electron counting camera (Gatan) positioned after a GIF BioQuantum energy filter (Gatan). Cryo-EM data were acquired using EPU software (FEI) at a nominal magnification of x105,000, with a pixel size of 0.85 or 0.87 Å per pixel. Movies of a total fluence of ~50 electrons per Å² were collected at ~1 e-/Å² per frame. A total number of 15,488 (APO-); 13,462 (DISTAL-PROMOTER and MID-LINK); 13,204 (PRE-INITIATION); 10,368 (ELONGATION) movies were acquired at a defocus range from −0.4 to −3.1 µm (Supplementary Table 2).

*3′ END- structures.* The grids were loaded into an FEI Tecnai Krios electron microscope at European Synchrotron Radiation Facility (ESRF) beamline CM01[49], operated at an accelerating voltage of 300 kV and equipped with K2 Summit direct electron counting camera (Gatan) positioned after a GIF Quantum energy filter (Gatan). Cryo-EM data were acquired using EPU software (FEI) at a nominal magnification of x165,000, with a pixel size of 0.827 Å per pixel. Movies of a total fluence of ~50 electrons per Å² were collected at ~1 e-/Å² per frame. A total

number of 6616 movies were acquired at a defocus range from −0.3 to −2.8 μm (Supplementary Table 2).

*Cryo-EM image processing.* All movie frames were aligned and dose-weighted using the MotionCor2 programme (Supplementary Figs. 24, 26, 28, 30, 32). Thon rings from summed power spectra of every 4 e⁻/Å² were used for contrast-transfer function parameter calculation with CTFFIND 4.1[50]. Particles were selected with WARP[51]. The further 2D and 3D cryo-EM image processing was performed in RELION 3.1[52]. First, particles were iteratively subjected to two rounds of 2D classification (Supplementary Figs. 24, 26, 28, 30, 32) at 2x binned pixel size. Particles in classes with poor structural features were removed.

*3D analysis of the APO-structures.* Two times binned particles (1491 k) were subjected to two rounds of 3D classifications with image alignment (Supplementary Fig. 25). The first round of 3D classification was restricted to ten classes and performed using a 60 Å low-pass filtered initial model constructed from 16 most populated 2Ds class averages (Supplementary Fig. 25). Particles in classes with poor structural features were removed. The second classification (into ten classes) was done during two rounds of 25 iterations each, using the regularisation parameter $T = 4$. In the second round, local angular searches were performed at 3.5° to clearly separate structural species. Three major 3D species were identified: the bare CORE-like, the ENDO-like and RIBBON-like.

In the first branch of 3D classification, the focus was on the structure of the CORE of the LASV L protein. All identified species were pooled together (428.8 k particles) and CORE-focused global 3D refinement was performed. Another round of CORE-focused 3D classification was performed based on the global refinement with local angular searches at 0.9° to clearly separate structural species. The most defined class (67.4 k particles) was further CORE-focused 3D auto-refined and iteratively aberration-corrected. For Bayesian-polishing only the first 23 frames were used.

In the second branch of 3D classification, the focus was on the structure of the ENDO-like species. The ENDO-like species from CSSB DATA 1 and DATA 2 were pooled together (209.4 k particles) and globally 3D auto-refined. A global 3D classification was performed based on the global refinement with local angular searches at 0.9°. The most defined class (74 k particles) was further core-focused 3D auto-refined as described for the CORE-like species.

In the third branch of 3D classification, the focus was on the structure of the RIBBON-like species (35.4 k particles). The RIBBON structure was obtained in a similar way as the ENDO one.

*3D analysis of the 3′ END structures.* Two times binned particles (654.8 k) were subjected to global 3D auto-refinement with 60 Å low-pass filtered APO-ENDO structure as the initial model. In the first branch of 3D classification, the focus was on the structure of the 3′ end binding site. A specific mask containing the 3′ vRNA secondary binding site and pyramid domain area was created (yellow, Supplementary Fig. 27) and focused 3D classification without angular assignment was performed, using regularisation parameter $T = 4$. The most defined class (194 k particles) was globally 3D auto-refined, iteratively aberration-corrected and Bayesian-polished (only the first 23 frames were used). The resulting global refinement was then subjected to core-focused 3D classification, with angular assignment using regularisation parameter $T = 8$. The most defined class (84.5 k particles) was CORE-focused 3D auto-refined.

In the second branch of 3D classification, the focus was on the structure of the ENDO-like species. A specific mask containing the ENDO area was created (cyan, Supplementary Fig. 27) and ENDO-focused 3D classification without angular assignment was performed, using regularisation parameter $T = 4$. The most defined class (159 k particles) was globally 3D auto-refined, iteratively aberration-corrected and Bayesian-polished (only the first 26 frames were used). The resulting global refinement was then subjected to ENDO-focused (cyan mask, Supplementary Fig. 28) 3D classification, without angular assignment using regularisation parameter $T = 12$. The most defined class (40.2 k particles) was globally 3D auto-refined.

*3D analysis of the DISTAL-PROMOTER and MID-LINK structures.* Two times binned particles (2,081 k) were subjected to two rounds of three-dimensional classifications with image alignment (Supplementary Fig. 31). The first round of 3D classification was restricted to 12 classes and performed using 60 Å low-pass filtered APO-ENDO structure as the initial model. Particles in classes with poor structural features were removed. The second classification (into ten classes) was done during two rounds of 25 iterations each, using the regularisation parameter $T = 4$. In the second round, local angular searches were performed at 3.5° to clearly separate structural species. Three major 3D species were identified: DISTAL-PROMOTER-like, the MID-LINK-like and ENDO-like. The ENDO-like particles were processed together with CSSB DATA 1 (Supplementary Fig. 25).

The DISTAL-PROMOTER-like and MID-LINK-like species were pooled (65 k particles) and globally 3D auto-refined. In the first branch of 3D classification, the focus was on the structure of the DISTAL-PROMOTER-like specie. A specific mask containing the DISTAL-PROMOTER area was created (pink, Supplementary Fig. 31) and DISTAL-PROMOTER -focused 3D classification without angular assignment was performed, using regularisation parameter $T = 4$. The most

defined class (23.7 k particles) was further DISTAL-PROMOTER -CORE-focused (purple mask, Supplementary Fig. 31) 3D auto-refined as described for the CORE-like specie.

In the second branch of 3D classification, the focus was on the structure of the MID-LINK-like specie. A specific mask containing the MID-LINK area was created (orange, Supplementary Fig. 31) and MID-LINK -focused 3D classification without angular assignment was performed, using regularisation parameter $T = 4$. The most defined class (40.7 k particles) was further MID-LINK-CORE-focused (yellow mask, Supplementary Fig. 31) 3D auto-refined as described for the CORE-like species.

*3D analysis of the PRE-INITIATION structure.* Particles (2,470 k) were subjected to two rounds of reference-free 2D classification. Particles in classes with secondary structure features were selected (1016 k particles) and used for an ab initio volume reconstruction and then 3D refined using the latter ab initio 60 Å low-pass filtered volume reconstruction as the initial model. The particles were astigmatism corrected with CTFrefine within RELION. The particles were then subjected to a 3D classification restricted to twelve classes using $T = 4$ and 7.5° sampling for 25 iterations and then 3.5° sampling for an additional ten iterations. Classes with comparable structural features were combined (319 k particles), 3D refined, aberration-corrected and Bayesian-polished then 3D refined again. The refined particles were then subjected to further 3D classification without image alignment and particles from the most defined class (119 k particles) were used for final 3D auto-refinement (Supplementary Figs. 28 and 29).

*3D analysis of the ELONGATION structure.* Particles (2452k) were binned four times and subjected to one round of reference-free 2D classification. Particles in classes with secondary structure features were selected (579 k particles) and subjected to 3D classification restricted to ten classes using $T = 4$ and 7.5° sampling for 25 iterations with the 60 Å low-pass filtered DISTAL-PROMOTER volume as a reference. Classes with comparable structural features were combined (122 k particles) and 3D refined to 3.5 Å resolution. The 20 Å low-pass filtered refined volume was then used as a reference for the 3D classification of all extracted particles (2452k) restricted to eight classes using $T = 4$ and 7.5° sampling for 35 iterations. The most defined class (426 k particles) was selected for 3D refinement and then subjected to further 3D classification without image alignment restricted to six classes. The most defined class (79 k particles) was 3D refined, aberration-corrected and Bayesian-polished then finally 3D refined using SIDESPLITTER[53] (Supplementary Figs. 32 and 33).

*Final steps.* All final cryo-EM density maps were generated by the post-processing feature in RELION and sharpened or blurred into MTZ format using CCP-EM[54]. The resolutions of the cryo-EM density maps were estimated at the 0.143 gold standard Fourier Shell Correlation (FSC) cut-off (Supplementary Table 2). A local resolution (Supplementary Table 2) was calculated using RELION and reference-based local amplitude scaling was performed by LocScale within CCP-EM[55].

*Model building.* The APO-CORE structure was constructed de novo with iterative rounds of model building with Coot[56] and real-space refinement with Phenix[57]. Subsequent structures used this as a basis for the further model extension. Secondary structure prediction using JPRED[58] based on multiple sequence alignment of both New World and Old World arenaviruses (Supplementary Alignment file) was particularly helpful in guiding model building. Considerable care was taken to cross-check between structures for consistency of sequence assignment and to ascertain connectivity. This also enabled better completion of models in lower resolution maps by transfer of structural elements that could be more accurately modelled in a higher resolution map. A homology model based on the MACV pendant domain (PDB: 6KLD), rebuilt to correct for sequence misalignments (using the original map, EMD-0707), was initially used to help build the LASV pendant domain. A 3D structure prediction using AlphaFold2[59] via the server https://colab.research.google.com/github/deepmind/alphafold/blob/main/notebooks/AlphaFold.ipynb resulted in an improved model for the pendant domain, which was used for model building. Unexpectedly, despite apparent sequence homologies, the LASV α-bundle 827-VVVNK…IIDQY-925 has a completely different arrangement of helices (topologically impossible to align in 3D) than that of the equivalent region in MACV L 820-VVIPK…QVALA-917 (Supplementary Fig. 6), both structures being confirmed by good quality maps. This might partly explain why in the LASV L structure previously published[9] (PDB:6KLC), based on a lower resolution 3.9 Å map, the α-bundle is built in the reverse direction.

Buried surface areas were determined using the Protein interfaces, surfaces and assemblies' service PISA at the European Bioinformatics Institute (http://www.ebi.ac.uk/pdbe/prot_int/pistart.html)[60]. An overview of the Segment-based Manders' Overlap Coefficient (SMOC) scores[61] for each of the structures is provided in Supplementary Fig. 34. Structure presentation was done using PyMOL (Schrödinger) and UCSF ChimeraX[62].

*Electrophoretic mobility shift assay.* 3′ RNA 1–10 nt (Supplementary Table 1) was chemically synthesised with a fluorophore at the 5′ end (5′ Cyanine3) (Biomers). Reactions containing 0–1 μM L protein and 0.2 μM labelled single-stranded RNA

were set up in binding buffer (100 mM HEPES(NaOH), pH 7.0, 100 mM NaCl, 50 mM KCl, 2 mM MnCl₂, 2 mM dithiothreitol, 0.1 μg/μL Poly(C) RNA (Sigma), 0.1 μg/μL bovine serum albumin and 0.5 U/μl RNasin (Promega)) and incubated on ice for 30 min. RNA-bound protein complexes were separated from unbound RNA by native gel electrophoresis at 4 °C, using 5% polyacrylamide Tris-glycine gels. Fluorescence signals were detected in the gel with the VILBER LOURMAT FUSION SL4 using the Starlight Module with an excitation wavelength of 523 and a 609 nm emission filter.

*Endonuclease assay*. An RNA 16-mer was chemically synthesised with either 5′ Cap (TriLink BioTechnologies) or 5′ Triphosphate (Chemgenes). For labelling pCp-Cy5 (Cytidine-5′-phosphate-3′-(6-aminohexyl)phosphate (Jena bioscience), was ligated to the 3′ end of the 16 nt RNA using T4 RNA ligase (Thermo Scientific). The resulting labelled 17 nt RNA substrates (Supplementary Table 1) were separated from excess pCp-Cy5 by denaturing PAGE (7 M urea, 25% acrylamide 0.5-fold Tris-borate-EDTA). The clearly blue coloured product bands were excised from the gel. The gel pieces were grounded and the RNA was extracted two times with Tris-borate buffer. The pure labelled RNA was precipitated with 90% Ethanol from the supernatant after the addition of Ammonium acetate (2.5 M), washed two times with 90% Ethanol and dissolved in DEPC treated H₂O. Reactions containing 0.5 μM L protein were incubated, sequentially with 2.5 pmol of either single-stranded 5′ promoter RNA, 3′ promoter RNA (Supplementary Table 1) or both, on ice for 15 min in 5 μl assay buffer (100 mM HEPES(NaOH) pH 7.0, 100 mM NaCl, 50 mM KCl, 2 mM MnCl₂, 0.5 U/μl RNasin (Promega), 2 mM dithiothreitol, and 0.1 μg/μL bovine serum albumin). After the addition of ~0.3 μM labelled RNA the mix was incubated at 37 °C for 120 min. The reaction was stopped by adding an equivalent volume of RNA loading buffer (98% formamide, 18 mM EDTA, 0.025 mM SDS) and heating the samples at 95 °C for 5 min. Products were separated by denaturing PAGE on 7 M Urea, 25% polyacrylamide Tris-borate-EDTA (0.5-fold) gels and 0.5-fold Tris-borate buffer. Fluorescence signals were detected in the gel with the VILBER LOURMAT FUSION SL4 using the Starlight Module with an excitation wavelength of 624 nm and a 695 nm emission filter. Uncropped images are provided as Source Data.

## Polymerase assay

*Standard polymerase assay*. If not indicated otherwise, 0.5 μM L protein was incubated sequentially with 1 μM of single-stranded 5′ promoter RNA (Supplementary Table 1) and 1 μM single-stranded 3′ promoter RNA (Supplementary Table 1) in assay buffer (100 mM HEPES(NaOH) pH 7.0, 50 mM NaCl, 50 mM KCl, 2 mM MnCl2, 0.5 U/μl RNasin (Promega), 2 mM dithiothreitol) on ice for 15 min. The reaction was started by the addition of NTPs (0.25 mM UTP/ATP/CTP and 0.125 mM GTP supplemented with 166 nM, 5 μCi [α]³²P-GTP) in a final reaction volume of 10 μL. After incubation at 30 °C for 2 h the reaction was stopped by adding an equivalent volume of RNA loading buffer (98% formamide, 18 mM EDTA, 0.025 mM SDS, xylene cyanol and bromophenol blue) and heating the sample at 95 °C for 5 min. Products were separated by native gel electrophoresis using 25% polyacrylamide 0.5-fold Tris-borate-EDTA gels and 0.5-fold Tris-borate running buffer. Signals were visualised by phosphor screen autoradiography using a Typhoon FLA-7000 phosphorimager (Fujifilm) and the respective FLA-7000 software. Uncropped images are provided as Source Data.

*Primer-dependent polymerase assay*. Primer GCG, C1, St1 and C8 (Supplementary Table 1) were chemically synthesised with 5′-hydroxy ends (Biomers), primer C8ppp (Supplementary Table 1) with 5′ Triphosphate modification (Chemgenes). An N⁷-MeGppp (Cap0) was introduced at the 5′ terminus of C8ppp using the ScriptCap m7G Capping System (CELLSCRIPT) with 1 nmol C8ppp oligo using the manufacturer's standard protocol. After the addition of Ammonium acetate (2.5 M) the capped RNA was precipitated with Ethanol (90%), washed two times with Ethanol (90%), dried and dissolved in DEPC treated H₂O. For primer-dependent reactions, 10 μM of the respective primer was added to LASV L bound to promoter RNA and the mix was again incubated on ice for 15 min. The reaction was started by the addition of NTPs (0.25 mM UTP/ATP/CTP and 0.125 mM GTP supplemented with 166 nM, 5 μCi [α]³²P-GTP) in a final reaction volume of 10 μL.

*LASV mini-replicon system*. The experiments were performed in the context of the T7 RNA polymerase-based LASV mini-replicon system[15,24,36]. L genes were amplified using mutagenic PCR from a pCITE2a-L template to either yield wild-type or mutated L gene expression cassettes. L gene PCR products were further gel-purified when additional unspecific bands were visible in agarose gels and quantified spectrophotometrically. BSR-T7/5 cells stably expressing T7 RNA polymerase[63] were transfected per well of a 24-well plate with 250 ng of mini-genome PCR product expressing Renilla luciferase (Ren-Luc), 250 ng of L gene PCR product, 250 ng of pCITE-NP expressing NP, and 10 ng of pCITE-FF-luc expressing firefly luciferase as an internal transfection control. At 24 h post transfection, either total cellular RNA was extracted for Northern blotting using an RNeasy Mini kit (Qiagen) or cells were lysed in 100 μL of passive lysis buffer (Promega) per well, and firefly luciferase and Ren-Luc activity were quantified using the dual-luciferase reporter assay system (Promega). Ren-Luc levels were corrected with the firefly luciferase levels (resulting in standardised relative light units [sRLU]) to compensate for differences in transfection efficiency or cell

density. Data were evaluated using Prism 7.0d and are always presented as the mean of a given amount (n) of biological replicates as well as the respective standard deviation (SD).

For Northern blot analysis, 750 ng of total cellular RNA was separated in a 1.5% agarose-formaldehyde gel and transferred onto a Roti®-Nylon plus membrane (pore size 0.45 μm, Carl Roth). After UV cross-linking and methylene blue staining to visualise 28 S rRNA the blots were hybridised with a ³²P-labelled riboprobe targeting the Ren-Luc gene. Transcripts of Ren-Luc genes and complementary replication intermediate RNA of the minigenome were visualised by autoradiography using a Typhoon FLA-7000 phosphorimager (Fujifilm) and the respective FLA-7000 software. Quantification of signals for antigenomic RNA and mRNA was done in ImageJ 2.0.0-rc-43/1.50e (Source file). To verify the expression of L protein mutants in BSR-T7/5 cells, the cells were transfected with 500 ng of PCR product expressing C-terminally 3xFLAG-tagged L protein mutants per well in a 24-well plate. Cells were additionally infected with modified vaccinia virus Ankara expressing T7 RNA polymerase (MVA-T7)[64] to boost the expression levels and thus facilitate detection by immunoblotting. After cell lysis and electrophoretic separation in a 3–8% Tris-acetate polyacrylamide gel, proteins were transferred to a nitrocellulose membrane (GE Healthcare). FLAG-tagged L protein mutants were detected using peroxidase-conjugated anti-FLAG M2 antibody (1:10,000) (A8592; Sigma-Aldrich) and bands were visualised by chemiluminescence using Super Signal West Femto substrate (Thermo Scientific) and a FUSION SL image acquisition system (Vilber Lourmat). All uncropped images are provided as Source Data.

**Statistics and reproducibility**. Unless stated otherwise in the figure legends all in vitro assays (electrophoretic mobility shift assay, polymerase assay, endonuclease assay), as well as Coomassie-stained SDS gels, have been repeated with similar results at least twice. The total number of micrographs included in each cryo-EM dataset is given in Supplementary Table 2, representative micrographs are shown in Supplementary Fig. 24, 26, 28, 30 and 32.

**Reporting Summary**. Further information on research design is available in the Nature Research Reporting Summary linked to this article.

## Data availability

Coordinates and structure factors or maps generated in this study have been deposited in the wwwPDB or EMDB: Apo-structure of Lassa virus L protein (well-resolved polymerase core) [APO-CORE] EMD-12807, PDB ID 7OCH Apo-structure of Lassa virus L protein (well-resolved endonuclease) [APO-ENDO] EMD-12860, PDB ID 7OE3 Apo-structure of Lassa virus L protein (well-resolved α-ribbon) [APO-RIBBON] EMD-12953, PDB ID 7OE7 Lassa virus L protein bound to 3′ promoter RNA (well-resolved polymerase core and 3′ RNA secondary binding site) [3END-CORE] EMD-12862, PDB ID 7OEA Lassa virus L protein bound to 3′ promoter RNA (well-resolved endonuclease) [3END-ENDO] EMD-12863, PDB ID 7OEB Lassa virus L protein in a pre-initiation conformation [PRE-INITIATION] EMD-12955, PDB ID 7OJL Lassa virus L protein with endonuclease and C-terminal domains in close proximity [MID-LINK] EMD-12861, PDB ID 7OJJ Lassa virus L protein bound to the distal promoter duplex [DISTAL-PROMOTER] EMD-12954, PDB ID 7OJK Lassa virus L protein in an elongation conformation [ELONGATION] EMD-12956, PDB ID 7OJN All other data supporting the findings of this study are available within the article, its supplementary information and source data files. Additional information, relevant data and unique biological materials will be available from the corresponding author upon reasonable request. Source data are provided with this paper.

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

## Acknowledgements

We acknowledge the European Synchrotron Radiation Facility for the provision of beam time on CM01 and we would like to thank Daouda A.K. Traore for assistance. Furthermore, we want to thank Sophia Reindl and Nadja Hüttmann for advice and technical

support in the early phases of this project; Michael Hons, Wojtek Galej and Erika Pellegrini for access to the Glacios at EMBL Grenoble; Carolin Seuring and Cornelia Cazey for access to Cryo-EM facility at CSSB; Aymeric Peuch and Wolfgang Lugmayr for help with using the joint EMBL-IBS and the CSSB partition on the DESY computer cluster. We acknowledge funding of this project by the Leibniz Association's Leibniz competition programme (grant K72/2017 to S.G., K.G. and S.C.), the Federal Ministry of Education and Research of Germany (grant 01KI2019 to M.R.), the Wilhelm und Maria Kirmser-Stiftung, the Alexander von Humboldt foundation (FRA 1200789 HFST-P to E.R.J.Q.) as well as the EMBL Interdisciplinary Postdocs (EI3POD) initiative co-funded by Marie Skłodowska-Curie (grant 664726 to T.K.). Part of this work was performed at the Cryo-EM Facility at CSSB, supported by the UHH and DFG (grants INST 152/772-1 and 774-1 to K.G.).

## Author contributions

T.K., D.V., E.R.J.Q., S.G., K.G., M.R. and S.C. conceived and supervised the project. C.B. and M.M. carried out cloning. D.V. and C.B. expressed and purified the proteins. T.K., D.V. and S.R.T. prepared the cryo-EM grids. T.K., S.R.T. and E.R.J.Q. collected and processed the cryo-EM data, D.V., H.M.W., M.R. and S.C. built and validated the models; D.V. and H.M.W. performed the in vitro experiments, M.M. performed the cell-based mini-genome experiments, T.K., D.V., H.M.W. and M.R. compiled the figures, M.R. and S.C. wrote the manuscript with input from all co-authors. The authors wish it to be known that M.R. and S.C., in addition to the joint supervision statement, have equally contributed to this manuscript.

## Funding

## Competing interests

The authors declare no competing interests.
