## [Peer Review File · Nature Communications]

Conformational changes in Lassa virus L protein associated with promoter binding and RNA synthesis activityREVIEWER COMMENTS

Reviewer #1 (Remarks to the Author):

This is an outstanding manuscript describing different structures of the Lassa virus (LASV) polymerase. LASV, like many of its close relatives, is a significant human pathogen, and a greater understanding of its RNA transcription/ replication machinery is valuable for antiviral drug development as well as for basic science. The LSV L polymerase protein is multifunctional and undergoes conformational changes as it performs different stages in RNA transcription. In this manuscript, Kouba and coworkers present the L protein in a variety of conformations associated with different stages of RNA synthesis and validate the models they present using site-directed mutagenesis and cell-based and biochemical assays. This work significantly extends and complements previous studies of the LASV L protein and adds to the growing body of literature that reveals the similarities and differences between the polymerases of segmented, negative strand RNA viruses.

The structural data are clearly presented both in the main manuscript, the supplementary figures and the accompanying videos and the experiments to test enzyme function are well-designed with appropriate controls and experimental replicates. The methods are clear and easy to follow. The main body of the manuscript is written with considerable detail, which will help those readers with very strong interest in this area, but also clearly conveys the major take-home messages from the work for readers who might not be so invested or familiar with the topic.

Overall, this is a highly impactful, beautifully presented manuscript. I have only minor editorial comments, as follows:

1. Throughout the manuscript, there are graphs presenting the data obtained from mini-replicon experiments. It would be helpful if the authors could indicate if the vertical lines show standard error or standard deviation. In addition, it might be better to represent each individual data point within each bar.
2. In Supplementary Figure 10b, the mini-replicon luciferase data show only a minor effect on gene expression (and this is noted by the authors on lines 241-242). However, the Northern blot data that are presented show that the F1592A substitution results in an increase in antigenome (with only a minor effect on mRNA). If this is the case, it would suggest that replication is augmented and transcription is inhibited (relative to the amount of replicative template). If this is a reproducible finding, it would be helpful if the authors could comment on this.
3. Lines 534-546 in the discussion describe primary data. This text might be better integrated into the results section, so that the discussion section can focus on the models and key take-home messages.

4. In Supplementary Table 2, it would be helpful to include the GCG primer used in Supplementary Figure 18, so that the 5' and 3' moieties are identified (as with the other primers).

Reviewer #2 (Remarks to the Author):

Kouba et al. described cryo-EM structures of Lassa virus L protein in the apo-, pre-initiation, and elongation states. With a total of nine structures, multiple conformational rearrangements of motifs and subdomains have been observed, although the atomic models are not 100% complete. In this manuscript, Kouba et al. revealed that the endonuclease domain is inhibited in the apo- and elongation-states, while the endonuclease domain is uninhibited in the pre-initiation state. Mutagenesis of the critical residues identified in the structures was also used to validate the functional roles of key motifs. Kouba et al. also compared it with the L protein from other viral L proteins, such as La Crosse, MACV L, and influenza virus polymerase, and highlighted the similarities and differences among them.

Overall, it is a paper with excellent experimental data. The structure of apo-Lassa L was previously determined, but no RNA-bound pre-initiation or elongation states have been determined. The reported structures showed the potential domain movements upon promoter binding and in different states.

However, there are still some issues of this manuscript that need to be addressed:

1. Line 140. “~90% of the residues”. Please list the exact residue ranges for the models for all nine structures, and a brief comparison will be better.
2. Lines 187-188. “Moreover, it is so far unclear whether any protein segment might serve as a priming loop”. LASV L does not do de novo RNA synthesis, and why a priming loop is needed?
3. Figure 1d. There are two double mutants N331A/K332A and Y1450A/R1452A showing no activity compared to WT, while all other single mutants do not. How about a single mutant of these residues?
4. Figure 1e. What about the gel shift effects of N331A/K332A?

5. Figure 3c. What are the RMSDs for those PA-like, PB1-like, and PB2-like domains compared to influenza PA, PB1, and PB2? Kouba et al. compared the elongation state. What about the apo and pre-initiation states? Any major differences?

6. Figure 4e and 4f. Please show the density for the UMPNPP and the metal ions.

7. Figure 5b. The conformational differences of the inhibitory peptide. It is a bit confusing about the inhibitory peptide in three different states. Please show 3END-CORE and ELONGATION domains the same as the MID-LINK. Alternatively, show MID-LINK the same as two other domains plus the superimposition of three.

8. Figure 7. The schematic diagram of conformational changes in L proteins upon promoter binding and RNA synthesis. There are a total of nine structural models presented in this manuscript, and it is not clear how those nice structures related to each other. Please highlight the major differences among them and their sequential order in the a, b, c.

9. Lines 538-541, 560-571. Kouba et al. claimed no difference detected in primed product for the uncap vs capped primers, and the true cap-dependent transcription is still unclear. Is it possible that a different cap form is needed rather than cap0 to have a higher activity to the complex?

Reviewer #1 (Remarks to the Author):

This is an outstanding manuscript describing different structures of the Lassa virus (LASV) polymerase. LASV, like many of its close relatives, is a significant human pathogen, and a greater understanding of its RNA transcription/ replication machinery is valuable for antiviral drug development as well as for basic science. The LSV L polymerase protein is multifunctional and undergoes conformational changes as it performs different stages in RNA transcription. In this manuscript, Kouba and coworkers present the L protein in a variety of conformations associated with different stages of RNA synthesis and validate the models they present using site-directed mutagenesis and cell-based and biochemical assays. This work significantly extends and complements previous studies of the LASV L protein and adds to the growing body of literature that reveals the similarities and differences between the polymerases of segmented, negative strand RNA viruses.

The structural data are clearly presented both in the main manuscript, the supplementary figures and the accompanying videos and the experiments to test enzyme function are well-designed with appropriate controls and experimental replicates. The methods are clear and easy to follow. The main body of the manuscript is written with considerable detail, which will help those readers with very strong interest in this area, but also clearly conveys the major take-home messages from the work for readers who might not be so invested or familiar with the topic.

Overall, this is a highly impactful, beautifully presented manuscript. I have only minor editorial comments, as follows:

1. Throughout the manuscript, there are graphs presenting the data obtained from mini-replicon experiments. It would be helpful if the authors could indicate if the vertical lines show standard error or standard deviation. In addition, it might be better to represent each individual data point within each bar.

We thank the reviewer for pointing this out. The vertical bars used indicate the standard deviation. The number of biological replicates is always at least 3, sometimes up to 7 allowing for a reliable assessment of mean average and standard deviation as a measure of variability. As requested, we included the single data points in all bar graphs. We also included a source file with all single data points.

2. In Supplementary Figure 10b, the mini-replicon luciferase data show only a minor effect on gene expression (and this is noted by the authors on lines 241-242). However, the Northern blot data that are presented show that the F1592A substitution results in an increase in antigenome (with only a minor effect on mRNA). If this is the case, it would suggest that replication is augmented and transcription is inhibited (relative to the amount of replicative template). If this is a reproducible finding, it would be helpful if the authors could comment on this.

Thank you for bringing this up. We quantify the Northern blot signals of all L mutants and calculate the transcription-to-replication signal ratio relative to the corresponding average wildtype signals. We usually include two wild-type RNA samples from separate cell culture wells in the same section on the Northern blot as the transcription-to-replication signal ratio also fluctuates a bit

among wild-type L proteins. The transcription-to-replication signal ratio determined for F1592A is 0.86. This means that relative to the transcription-to-replication signal ratio in the wild-type L lanes (set to 1) there is a very slight reduction of transcription relative to replication in this mutant. However, this is still within the range of normal fluctuation based on our experience from numerous previous studies with the LASV minireplicon system (Reguera et al. Plos Pathog 2016; Lehmann et al. JVI 2014; Hass et al. JVI 2008; Lelke JVI 2010; Hass et al. JVI 2004; list not exhaustive). Therefore, we always rely on several measures to determine a selective transcription defect, which are a strong reduction Luciferase activity to $\leq 25\%$ of wild-type L protein activity, a wild-type like antigenome synthesis level ($>35\%$) and a clear reduction of the mRNA-to-antigenome (transcription-to-replication) ratio to ≤ 0.5 . In contrast to mutant F1592A, the endonuclease active site mutant D89A (Supplementary Fig. 12) shows a clear selective transcription defect: transcription-to-replication signal ratio 0.45, Luciferase activity 0.68% and antigenome level 37.9%. This is not the case for mutant F1592A, which indeed shows a robust signal of $\sim 60\%$ of wildtype luciferase activity and we therefore do not consider this a selective transcription defect.

For transparency, we included a source file with a table listing the determined luciferase activity values as well as Northern blot quantification results for all mutants. Additionally, we provide a file containing all uncropped gels/blots etc.

3. Lines 534-546 in the discussion describe primary data. This text might be better integrated into the results section, so that the discussion section can focus on the models and key take-home messages.

This mentioned paragraph (now lines 526-537) describes experimental data directly relevant to the discussion as these experiments address somewhat controversial results, which would need additional explanation of the rationale in the results section. We feel that they make most sense in context with the described results of others and guiding towards our interpretation of our structural and functional data. We therefore would like to keep them in the discussion.

4. In Supplementary Table 2, it would be helpful to include the GCG primer used in Supplementary Figure 18, so that the 5' and 3' moieties are identified (as with the other primers).

Done.

Reviewer #2 (Remarks to the Author):

Kouba et al. described cryo-EM structures of Lassa virus L protein in the apo-, pre-initiation, and elongation states. With a total of nine structures, multiple conformational rearrangements of motifs and subdomains have been observed, although the atomic models are not 100% complete. In this manuscript, Kouba et al. revealed that the endonuclease domain is inhibited in the apo- and elongation- states, while the endonuclease domain is uninhibited in the pre-initiation state. Mutagenesis of the critical residues identified in the structures was also used to validate the functional roles of key motifs. Kouba et al. also compared it with the L protein from other viral L proteins, such as La Crosse, MACV L, and influenza virus polymerase, and highlighted the similarities and differences among them.

Overall, it is a paper with excellent experimental data. The structure of apo-Lassa L was previously determined, but no RNA-bound pre-initiation or elongation states have been determined. The

reported structures showed the potential domain movements upon promoter binding and in different

states

However, there are still some issues of this manuscript that need to be addressed:

1. Line 140. “~90% of the residues”. Please list the exact residue ranges for the models for all nine structures, and a brief comparison will be better.

We added a table listing all residue ranges included in all LASV L protein models as Supplementary Table 3. This is also visible in Supplementary Fig. 24, a comparison of all models by TEMPy SMOC scores, although the exact amino acid ranges can't be identified there.

2. Lines 187-188. “Moreover, it is so far unclear whether any protein segment might serve as a priming loop”. LASV L does not do de novo RNA synthesis, and why a priming loop is needed?

Bunyaviruses use different mechanisms for genome transcription and genome replication. According to sequencing data (Auperin et al. JVI 1984; Raju et al. 1990 Virology; Polyak et al. 1995 JVI; Garcin et al. JVI 1992; Auperin et al. Virology 1986; Garcin et al. JVI 1995) and biochemical studies (Vogel et al. JBC 2019), LASV (and other bunyaviruses) uses a primer, generated by cap-snatching, to initiate viral transcription. However, it most likely initiates genome replication de novo by using a prime-and-realign mechanism. This is mentioned in the introduction lines 67-68 “Viral genome replication is initiated by a prime-and-realign mechanism resulting in an extra G nucleotide at the 5' end of the vRNA and cRNA”. It is expected that a protein component is needed for placement of the first nucleotide for catalysis of the initial phosphodiester bond. This protein residue (or these residues) might be placed in a so-called priming-loop as observed for influenza virus, but it might be also within other structural elements of L. As there has been speculation about the priming loop of bunyavirus L proteins already from apo-structures we felt it was important to emphasize that we cannot and do not want to draw conclusions on the priming loop based on our structures.

3. Figure 1d. There are two double mutants N331A/K332A and Y1450A/R1452A showing no activity compared to WT, while all other single mutants do not. How about a single mutant of these residues?

In the secondary 3' end binding site there are multiple protein residues involved in specific binding of 7 RNA nucleotides. The minigenome experiments were performed in order to investigate if the integrity of this binding site is important for L protein function. Due to the high number of contacts, it is not surprising that double mutants show a more pronounced effect than single site mutants. The purpose of these experiments was not to pinpoint specific residues of the secondary site that are especially or selectively important for L protein function but to investigate the overall effect on L protein function. We have backed up these functional data with gel shift assays demonstrating that the inactive double mutant Y1450A/R1452A of the L protein shows severely reduced binding to 3' RNA. We therefore do not think that investigating further single site mutants in the minigenome system would provide relevant insights into the general functional importance of this complex RNA binding site.

4. Figure 1e. What about the gel shift effects of N331A/K332A?

Production of a mutant full-length L protein for testing in the EMSA requires a long process including cloning of a 7 kb L protein mutant, production of the respective baculovirus, time-course experiments for expression kinetics, larger-scale expression and protein purification. Of course, one can always test more mutants but as the gel shift assays were used to complement the structural and functional minigenome data, we do not think it justifies the resources to show this again with a second mutant if the conclusion overall is plausible based on three different experimental approaches.

5. Figure 3c. What are the RMSDs for those PA-like, PB1-like, and PB2-like domains compared to influenza PA, PB1, and PB2? Kouba et al. compared the elongation state. What about the apo and pre-initiation states? Any major differences?

There is considerable evolutionary distance between Lassa virus L and influenza virus polymerase complex, which translates into structural divergence. In this context, we think it is more useful to focus on architectural similarities and differences rather than citing RMSD values domain by domain, which would require careful definition of what regions to superpose and what to exclude, something that could perhaps be done better in a Review comparing all such polymerases.

6. Figure 4e and 4f. Please show the density for the UMPNPP and the metal ions.

As suggested by the reviewer, we included the density for UMPNPP and the two metal ions in figures 4e and 4f. The figure and caption have been adapted accordingly.

7. Figure 5b. The conformational differences of the inhibitory peptide. It is a bit confusing about the inhibitory peptide in three different states. Please show 3END-CORE and ELONGATION domains the same as the MID-LINK. Alternatively, show MID-LINK the same as two other domains plus the superimposition of three.

We agree with the reviewer that this is a complex comparison. In the process of preparing this manuscript we have re-done this figure several times in order to make it most easy to understand for the readers. In this figure the EN is shown in the same orientation both for the 3END-CORE and ELONGATION structure. For the MID-LINK the active site is not occupied and overall in the same conformation as shown for 3END-CORE (as pointed out in the figure caption), we therefore removed the third structure from the superimposition as it only adds noise without containing important information. Supplementary Figure 11a complements this figure by showing the different positions of the inhibitory peptide from the perspective of this peptide (keeping a similar view on the peptide in this case). We added a sentence referring to Supplementary Figure 11 to the caption of Figure 5.

8. Figure 7. The schematic diagram of conformational changes in L proteins upon promoter binding and RNA synthesis. There are a total of nine structural models presented in this manuscript, and it is not clear how those nice structures related to each other. Please highlight the major differences among them and their sequential order in the a, b, c.

We have changed Figure 7 according to the reviewers' suggestion. The schematic of the structural rearrangements in the L protein now includes the information which structures the panels a, b and c are based on. Although we describe the conformational changes in this figure, we did not indicate any assignment of a sequential order of the conformational changes as this would be speculative at this stage. We amended the caption of Figure 7 for a more detailed description of conformational changes happening and to define which information is yet missing.

9. Lines 538-541, 560-571. Kouba et al. claimed no difference detected in primed product for the uncapped vs capped primers, and the true cap-dependent transcription is still unclear. Is it possible that a different cap form is needed rather than cap0 to have a higher activity to the complex?

We thank the reviewer for this interesting question. It is possible, that Lassa virus L protein accesses specific mRNA pools with further modifications than methylation at the 5'G N7, which would be the classical cap0, but so far there are no sound hypotheses about this topic. mRNA can be modified in various ways (see here for an overview: [10.1371/journal.pone.0102895](https://doi.org/10.1371/journal.pone.0102895) or [10.1038/s12276-020-0407-z](https://doi.org/10.1038/s12276-020-0407-z)) and it is outside the scope of this manuscript to screen different mRNA modifications and sequences. Although we expected the L protein to have at least some preference for a capped vs. uncapped RNA, as was claimed by others before, we were ourselves surprised to see that this is not the case and repeated the experiment several times. However, this finding matches with the fact that the isolated C-terminal domain of Lassa virus L protein, containing the CBD-like domain, was found unable to bind to m7GTP (cap0 analogue) in vitro (Lehmann et al. 2014), whereas phenuivirus CBDs showed binding to m7GTP in different experimental setups (Gogrefe et al. 2019, Vogel et al. 2020).

To address the reviewers' comment we again repeated the experiment including both cap0 and cap1 primers but could still not detect any difference in comparison to uncapped primers (OH or PPP termini). We added this new gel image to Supplementary Figure 22. This experiment does not exclude that other RNA modifications might be important, but the lack of a preference for capped vs. uncapped primers hints towards mechanistic differences between the bunyavirus families and is thus interesting.

REVIEWERS' COMMENTS

Reviewer #1 (Remarks to the Author):

The authors have carefully addressed the comments of both reviewers. They have provided sound explanations in cases where they disagree with the reviewers' recommendations. In addition, they've considered new papers, that describe L-Z complex structures, which were published while this manuscript was under review and have added a nice explanation of how their own data fits with the L-Z protein complex structure. Therefore, all concerns have been well addressed and the manuscript represents a strong and timely addition to the field.

Reviewer #2 (Remarks to the Author):

I want to thank the authors for addressing the initial comments. Following the revision to the article, I do not have more questions now.